# SCENT: Robust Spatiotemporal Learning for Continuous Scientific Data via Scalable Conditioned Neural Fields

**David Keetae Park** [1]  **Xihaier Luo** [1]  **Guang Zhao** [1]  **Seungjun Lee** [1]  **Miruna Oprescu** [2]  **Shinjae Yoo** [1]

## Abstract

Spatiotemporal learning is challenging due to the intricate interplay between spatial and temporal dependencies, the high dimensionality of the data, and scalability constraints. These challenges are further amplified in scientific domains, where data is often irregularly distributed (e.g., missing values from sensor failures) and high-volume (e.g., high-fidelity simulations), posing additional computational and modeling difficulties. In this paper, we present SCENT, a novel framework for scalable and continuity-informed spatiotemporal representation learning. SCENT unifies interpolation, reconstruction, and forecasting within a single architecture. Built on a transformer-based encoder-processor-decoder backbone, SCENT introduces learnable queries to enhance generalization and a query-wise cross-attention mechanism to effectively capture multi-scale dependencies. To ensure scalability in both data size and model complexity, we incorporate a sparse attention mechanism, enabling flexible output representations and efficient evaluation at arbitrary resolutions. We validate SCENT through extensive simulations and real-world experiments, demonstrating state-of-the-art performance across multiple challenging tasks while achieving superior scalability.

## 1. Introduction

Spatiotemporal learning focuses on modeling and interpreting data that exhibit variations in both space and time. This approach is crucial for analyzing intricate real-world phenomena where spatial structures are inextricably linked with temporal dynamics, including applications such as climate modeling (Reichstein et al., 2019), traffic forecasting (Li et al., 2018), medical imaging (Litjens et al., 2017), and video analysis (Wang et al., 2020). Achieving accurate spatiotemporal learning, however, presents significant challenges due to the presence of complex spatial-temporal dependencies, including spatial heterogeneity and temporal non-stationarity, compounded by the high dimensionality of the data. These challenges are further amplified in scientific and engineering domains, where datasets are frequently characterized by irregular distributions (e.g., arising from sensor malfunctions) and large volumes (e.g., generated by high-fidelity simulations), introducing further complexities in both computation and modeling.

To address the aforementioned challenges, significant research efforts have been dedicated to developing solutions from both traditional signal processing and, more recently, machine learning perspectives. Among the emerging machine learning approaches, Implicit Neural Representations (INRs) have garnered increasing attention due to their inherent flexibility (Sitzmann et al., 2020b; Mildenhall et al., 2020). INRs parameterize data as a continuous function, mapping coordinates to signal values (e.g., $(x, y) \rightarrow (r, g, b)$ for images), using a neural network. Despite their potential, the adoption of INRs in scientific domains has been limited. This can primarily be attributed to two key challenges: scalability and generalizability.

To bridge this methodological gap, we propose **SCENT** (**S**calable **C**ondition**e**d **N**eural Field for Spatio**T**emporal Learning), a novel framework designed to address the limitations of existing INR approaches. SCENT leverages a Transformer-based encoder-processor-decoder architecture to efficiently process large-volume spatiotemporal data, demonstrating strong scalability with respect to both dataset size and model parameter count. Furthermore, SCENT incorporates trainable query mechanisms to enhance generalizability, circumventing the computational overhead associated with existing strategies such as latent optimization (Dupont et al., 2022) or meta-learning (Chen & Wang, 2022).

Overall, the major contributions of our work include:

- We offer a unified framework for spatiotemporal learning, capable of performing joint interpolation, reconstruction, and forecasting.

---
[1]Computing and Data Sciences, Brookhaven National Laboratory [2]Cornell University, Cornell Tech. Correspondence to: David Keetae Park <dpark1@bnl.gov>.

*Proceedings of the 42nd International Conference on Machine Learning*, Vancouver, Canada. PMLR 267, 2025. Copyright 2025 by the author(s).

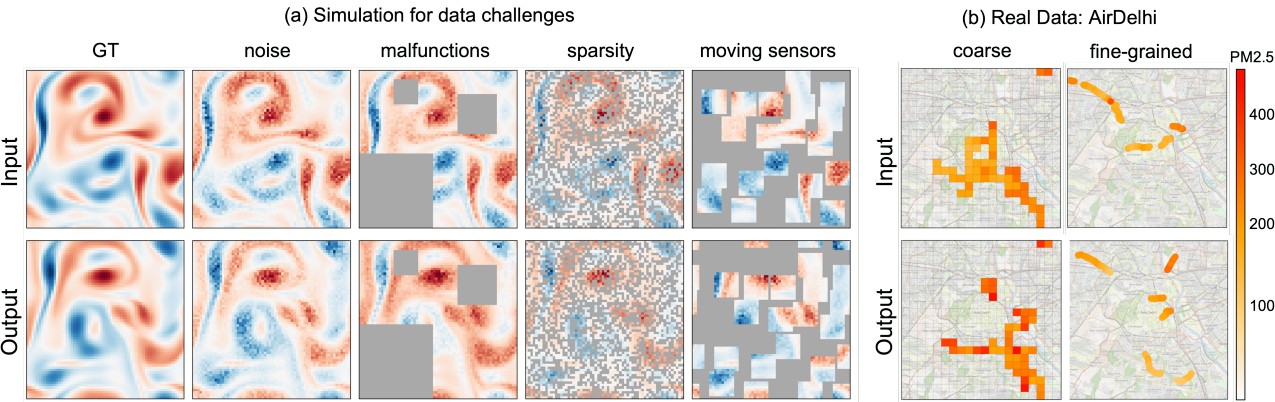

*Figure 1.* **Challenging data scenarios motivating this study.** (a) Learning continuous ground truth signal (GT) given noisy, malfunctioning, sparse, or moving sensors is a daunting challenge. However, these challenges are common in scientific data, including AirDelhi (Chauhan et al., 2024) data which measure the particulate matter levels (PM2.5) from moving vehicles.

- We empirically demonstrate that SCENT is scalable with larger models and datasets.
- We conduct extensive experiments to evaluate the efficacy of the proposed SCENT framework, demonstrating its superiority over state-of-the-art (SOTA) methods in a range of spatiotemporal learning tasks.

## 2. Problem Setting

**Background.** Spatiotemporal learning is a critical field for modeling data that evolves across space and time, with diverse applications in climate modeling (Reichstein et al., 2019), medical imaging (Litjens et al., 2017), and biochemistry (Jumper et al., 2021). While traditional methods like RNNs and LSTMs (Hochreiter & Schmidhuber, 1997) address temporal sequences, they often struggle with long-range dependencies and spatial correlations, limitations that Transformer-based architectures like ViT (Dosovitskiy et al., 2021) and TimeSformer (Bertasius et al., 2021) have begun to overcome. Implicit Neural Representations (INRs) offer a powerful, continuous approach to data modeling, mapping coordinates to function values using a neural network for compact encoding and resolution-agnostic interpolation (Sitzmann et al., 2020b; Mildenhall et al., 2020). However, standard INRs lack generalizability, requiring retraining for new data instances; this has led to the development of Generalizable INRs (GINRs) leveraging meta-learning (Sitzmann et al., 2020a; Chen et al., 2021; Dupont et al., 2022; Bauer et al., 2023). Conditioned Neural Fields (CNFs) are closely related to GINRs, providing a powerful, single-step framework for solving PDEs by learning continuous function representations conditioned on input parameters, aligning with GINRs' goal of learning continuous functions and addressing challenges in high-dimensional and sparse-data problems (Wang et al., 2024; Li et al., 2022). A detailed literature review is provided in the Appendix A.

**Problem Statement.** We consider a spatiotemporal discrete observation denoted as $u_{\mathbf{x}}^t$, where $\mathbf{x} = (x, y, z) \in \mathbb{R}^3$ represents the spatial coordinates, and $t \in \mathbb{R}$ denotes the sampled time. Given a set of $N_{\mathrm{i}}$ input observations measured at time $t_{\mathrm{i}}$, we define:

$$U^{t_{\mathrm{i}}} = \left\{ u_{\mathbf{x}_1}^{t_{\mathrm{i}}}, u_{\mathbf{x}_2}^{t_{\mathrm{i}}}, \dots, u_{\mathbf{x}_{N_{\mathrm{i}}}}^{t_{\mathrm{i}}} \right\}.$$

Our objective is to develop a model $F$ capable of learning the underlying continuous spatiotemporal function from these discrete $N_{\mathrm{i}}$ observations, enabling it to predict $M$ target outputs at a different time $t_{\mathrm{o}}$:

$$\hat{U}^{t_{\mathrm{o}}} = F\left(U^{t_{\mathrm{i}}}\right),$$

where the target outputs are given by:

$$U^{t_{\mathrm{o}}} = \left\{ u_{\mathbf{x}_1'}^{t_{\mathrm{o}}}, u_{\mathbf{x}_2'}^{t_{\mathrm{o}}}, \dots, u_{\mathbf{x}_{N_{\mathrm{o}}}'}^{t_{\mathrm{o}}} \right\}.$$

The specific task performed by $(F)$ is determined by the relationship between the input and output spatiotemporal coordinates:

- **Reconstruction.** If $\left(\{\mathbf{x}_j\}_{j=1}^{N_{\mathrm{i}}}, t_{\mathrm{i}}\right) = \left(\{\mathbf{x}_k'\}_{k=1}^{N_{\mathrm{o}}}, t_{\mathrm{o}}\right)$, the model is tasked with reconstructing the same observations it was provided.
- **Spatiotemporal Interpolation.** If $\left(\{\mathbf{x}_j\}_{j=1}^{N_{\mathrm{i}}}, t_{\mathrm{i}}\right) \neq \left(\{\mathbf{x}_k'\}_{k=1}^{N_{\mathrm{o}}}, t_{\mathrm{o}}\right)$, the model estimates the function at novel spatiotemporal locations.
- **Forecasting.** A special case of interpolation where $t_{\mathrm{o}} > t_{\mathrm{i}}$, requiring the model to predict future values.

Our goal is to develop a model $(F)$ capable of jointly performing reconstruction, interpolation, and forecasting of complex scientific data given arbitrary spatiotemporal coordinates $(x, y, z, t)$.

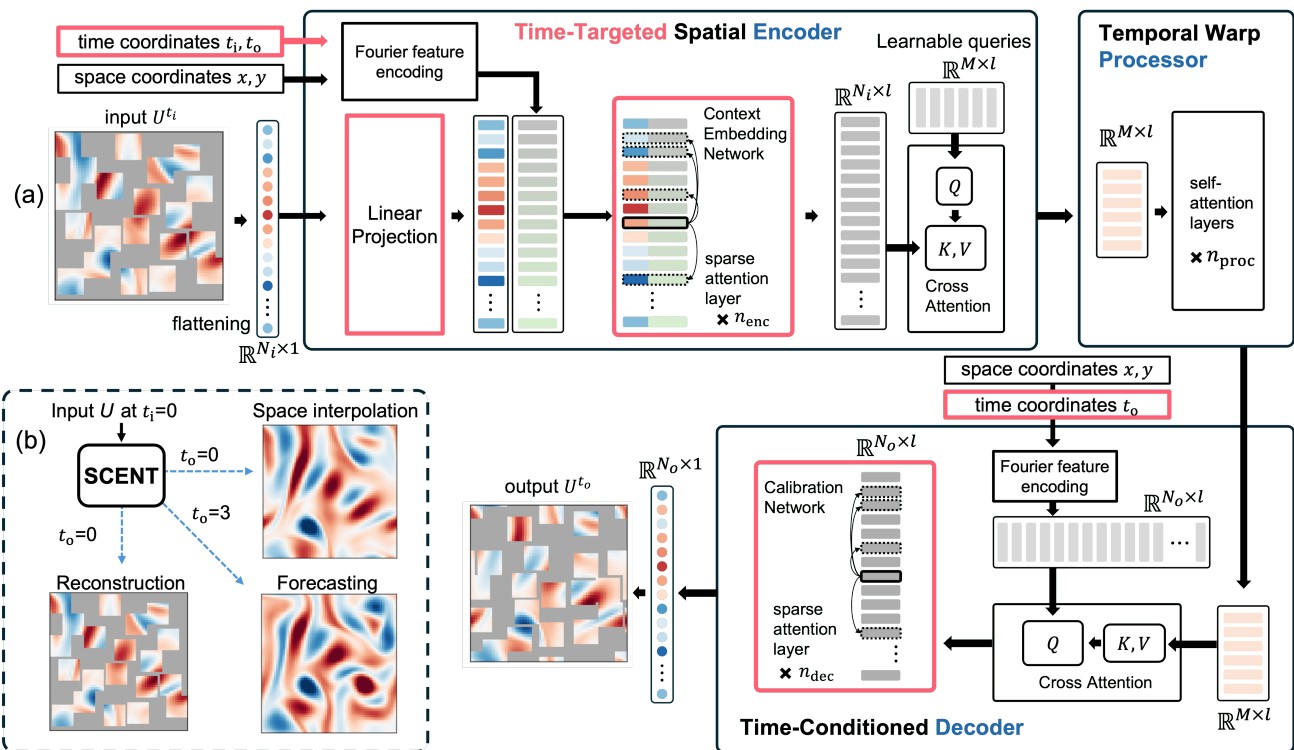

*Figure 2.* **SCENT overview.** (a) Detailed architecture of SCENT is illustrated. Our unique contributions are drawn with red boxes. Specifically, we introduce time coordinates to both encoder and decoder for learning continuous time representations. Also, we introduce Context Embedding Network and Calibration Network for improved spatial encoding and decoding, respectively. $n_{enc}$, $n_{proc}$, and $n_{dec}$ denote the number of layers in encoder, processor, and decoder respectively. (b) At the inference stage, a single SCENT model is capable to jointly perform reconstruction, spatiotemporal interpolation, and forecasting.

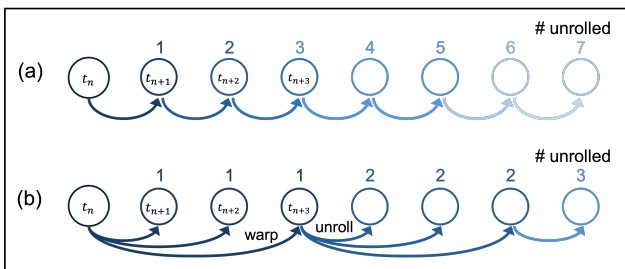

*Figure 3.* **Warp-unrolling forecasting (WUF).** (a) Conventional forecasting includes single-step unrolling which accumulates error. A dimming blue color is used to represent the increasing error. (b) WUF helps mitigate error accumulations caused by extensive unrolling steps.

# 3. SCENT

## 3.1. Encoder-Processor-Decoder Framework

The encoder-processor-decoder architecture is proposed as a general framework that scales linearly with input and output sizes, overcoming the quadratic scaling limitations of transformers with input sequence length (Jaegle et al., 2022). Prior works have leveraged this architecture to learn conditioned neural fields (CNFs) for solving partial differential equations (Lee, 2022; Lee & Oh, 2024; Serrano et al., 2024; 2023; Li et al., 2023), where the architecture is particularly beneficial, as these models flatten the input and treat each sample point as an individual token. Additionally, this architecture facilitates learning continuity-informed representations using inducing point learning within cross-attention layers (Jaegle et al., 2022; Lee & Oh, 2024). Building on these advancements, we evaluate the models' potentials in complex real-world scientific problems, often characterized by intricate noise patterns and irregular sensor distributions. In the remaining subsections, we describe notable developments on top of the existing architecture, as displayed by red-colored objects in Fig. 2(a).

## 3.2. Time-Targeted Spatial Encoder

The encoder processes an input data $U^{t_i} = \left\{ u^{t_i}_{\mathbf{x_1}}, u^{t_i}_{\mathbf{x_2}}, \ldots, u^{t_i}_{\mathbf{x_{N_i}}} \right\}$, representing $N_i$ samples from the space coordinate $\{\mathbf{x}_j\}_{j=1}^{N_i}$ at a given time $t_i$. Here, $\{\mathbf{x}_j\}_{j=1}^{N_i}$ can be structured on a grid or an irregular mesh. Using a cross-attention mechanism, the encoder transforms $U^{t_i}$ into a fixed-size set of tokens $Z^{t_i}_M = \{z_1, z_2, \ldots, z_M\}$, where $M$ is the number of latent trainable query tokens.

The encoder $E_\theta$ is then expressed as:

$$E_\theta : \left( U^{t_i}, \{\mathbf{x}_j\}_{j=1}^{N_i}, t_i, t_o \right) \to Z_M^{t_i},$$

where $t_i$ is the input time, and $t_o$ is the targeted output time. Including $t_o$ enhances attention to relevant information, leading to improved performance (Table 3).

Specifically, we encode the spatiotemporal coordinates $\{\mathbf{x}_j\}_{j=1}^{N_i}, t_i, t_o$ using Fourier features (Tancik et al., 2020). Fourier features map input coordinates to a higher-dimensional space using sinusoidal functions of varying frequencies, enabling models to capture fine-grained and periodic variations in the data. Meanwhile, the input samples $u_\mathbf{x}^{t_i}$ are separately embedded using a linear projection layer with parameters frozen. This effectively increases the dimensionality of function value representation $u_\mathbf{x}^{t_i}$, empirically enhancing performance. Both Fourier features and encoded samples are concatenated, and sent through Context Embedding Network (CEN) which consists of sparse self-attention layers to enrich context encoded in individual tokens. Within CEN, each token attends to randomly subsampled $S$ tokens where $S \ll N_i$. As the underlying data is in continuous field, sparse attention is an efficient way to encode global contexts to individual tokens. The final latent representation is then summarized through a cross attention against the learning queries which consist of $M$ tokens, where $M \ll N_i$. We justify the benefits of the linear projection and CEN by the ablation study (Table 3).

### 3.3. Temporal Warp Processor

We define the Time Warp Processor (TWP) for learning continuous temporal dynamics, denoted as $P_\theta : Z_M^{t_i} \to Z_M^{t_o}$, where $t_o = t_i + \Delta t, \Delta t \in [0, t_h]$ and $t_h$ represents a hyperparameter for the maximal time horizon used during training. Depending on $\Delta t$, TWP can either perform input reconstruction ($\Delta t = 0$) or forecasting ($\Delta t > 0$), allowing joint reconstruction and forecasting with a single model. This flexibility also enables the use of input-output pairs sampled at *non-integer* time intervals, which is particularly useful for spatiotemporal data with time-varying sampling rates (Chauhan et al., 2024; Nie et al., 2024).

**Novel Unrolling Strategy.** TWP can be leveraged to minimize accumulating prediction errors during long-horizon forecasting, which empirically leads to improved performance over baselines (Section 4.5). Previous models have been reported to struggle with long-term forecasting due to rapidly accumulating errors during inference and unrolling (Serrano et al., 2024). While we adhere to the standard practice of training models for next-state prediction, we do not perform one-step unrolling for the entire forecasting time steps. Instead, we utilize time warping, advancing directly to the time horizon $t_h$ once $t_o - t_i > t_h$, using

it as a reference state for predicting subsequent time steps (Fig. 3). This ensures that at any given time state, only a minimal number of prediction steps are required, thereby reducing the potential accumulation of errors. We denote this strategy as *warp-unrolling forecasting (WUF)* and use this for benchmark datasets (Table 2).

### 3.4. Time-Conditioned Decoder

The decoder $D_\theta : Z_M^{t_o} \to U^{t_o}$ utilizes the latent processed tokens $Z(t_o)$ to approximate the function values at $t_o$. To this end, we apply Fourier features encoding to $\mathbf{x}$ and $t_o$, and use the resulting queries for cross-attention against $Z(t_o)$. Then we apply $n_\text{dec}$ sparse self-attention layers (Calibration Network, abbreviated as CN) to calibrate spatiotemporal decoding. During training $U^{t_o}$ includes $N_o$ available samples at $t_o$. During inference, however, $Z(t_o)$ may be evaluated on arbitrary points in $\mathbf{x}$.

## 4. Experiments

We conduct evaluations using a diverse set of baseline models, encompassing state-of-the-art regular-grid methods such as FNO (Li et al., 2020), adaptable transformer architectures represented by OFormer (Li et al., 2023), as well as neural field-based approaches like DINO (Yin et al., 2023), CORAL (Serrano et al., 2023) and AROMA (Serrano et al., 2024). All training and evaluations are conducted using mean squared error (MSE), relative MSE (Rel-MSE), and root MSE (RMSE). RMSE is specifically used to measure PM2.5 levels for AirDelhi datasets (Section 4.1.3). Rel-MSE is defined as:

$$\text{Rel-MSE} = \frac{\sum_{j=1}^{N}(\hat{U}_j^{t_o} - U_j^{t_o})^2}{\sum_{j=1}^{N}(U_j^{t_o})^2}. \quad (1)$$

Most experiments were performed on a single NVIDIA H100 80GB HBM3 GPU. The largest model variant used in the scalability evaluation (Fig. 4) required distributed training across eight of them. Details of the algorithm can be found in Appendix B, while information on the dataset and training procedures is provided in Appendices C through G.

### 4.1. Datasets

#### 4.1.1. BENCHMARK NAVIER-STOKES DATASETS

We use three benchmark Navier-Stokes datasets (Li et al., 2020), each corresponding to different viscosity coefficients. These are designed to model the dynamics of a viscous and incompressible fluid governed by the 2D Navier-Stokes equation in vorticity form on the unit torus. The Navier-Stokes $1 \times 10^{-3}$ (NS-3) dataset (Yin et al., 2023; Serrano et al., 2023), with a viscosity coefficient $\nu = 1 \times 10^{-3}$, includes 256 training trajectories and 32 testing trajectories. NS-3 models relatively slower fluid dynamics with a time

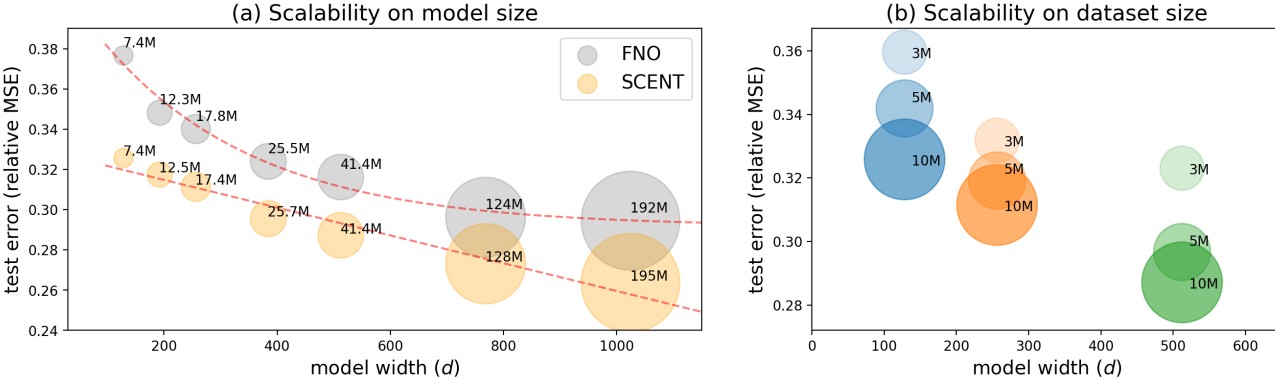

*Figure 4.* **Scalability evaluations.** (a) Texts next to each circle are the number of model parameters, and circle size is also proportional to it. Red dotted lines are scalability trends derived with exponential functions for comparisons. (b) Colors indicate training runs with identical model sizes. Texts next to each circle are the number of training instances, and the circle size is also proportional to it.

horizon of first 20 steps beyond the initial condition. The Navier-Stokes $1 \times 10^{-4}$ (NS-4) dataset, with a viscosity coefficient $\nu = 1 \times 10^{-4}$, is a more turbulent variant. Lastly, the Navier-Stokes $1 \times 10^{-5}$ (NS-5) dataset, with the lowest viscosity coefficient $\nu = 1 \times 10^{-5}$, represents the most turbulent fluid dynamics. Both NS-4 and NS-5 contain 1000 training and 200 testing trajectories and a maximum time steps of 30 and 20 steps, respectively. For all three datasets, we use the vorticity at $t_0 = 10$ as the initial condition for testing. These datasets provide a range of viscosity conditions, making them suitable for studying fluid dynamics across different turbulence levels.

### 4.1.2. Simulated Large-scale Complex Datasets

We introduce a new class of datasets to specifically simulate complex yet common real-world data scenarios as visualized in Fig. 1(a). This simulated dataset, derived from the Navier-Stokes equations, is designed to be significantly larger than benchmark datasets (Section 4.1.1), incorporating real-world challenges such as sensor noise, missing values, and dynamically moving or reconstructed sensor locations.

Concretely, we introduce five datasets. While each variant represents different data challenges, they consist of the same underlying ground truth. • **Ground truth** (S1): We utilize the Navier-Stokes equation with viscosity and boundary conditions to simulate highly complex and fast diffusion dynamics (Appendix E). This setup ensures that our simulations capture the intricate interactions of fluid motion, enabling a more realistic representation of challenging spatiotemporal processes. • **Noisy sensors** (S2): multiplicative noise is sampled (Nartasilpa et al., 2016) and applied to S1 following: $u'(\mathbf{x}, t) = u(\mathbf{x}, t) \cdot \eta(\mathbf{x}, t)$, where $\eta(\mathbf{x}, t) \sim \mathcal{N}(\mu, \sigma)$, $u$ is an input sample, $\eta$ is sample-level noise, and $\mathbf{x} = (x, y, z)$ and $t$ denote the spatial and temporal coordinates respectively. We use $\mu = 1$ and $\sigma = 0.2$. • **Sensor malfunctions** (S3): scientific sensors are often temporarily unavailable or

malfunctioning (Nathaniel et al., 2023). We empty out three square areas with different sizes to simulate lost sensors. • **Randomly sparse sensors** (S4): sensor locations may be randomly placed, such as for remote sensing (Myneni et al., 2001) in climate science. We randomly mask out 50% of the data to simulate the sparcity. • **Dynamically moving sensors** (S5): A more challenging scenario arises when sampling locations are sparse and dynamically shifting (Chauhan et al., 2024; Nie et al., 2024). To simulate this, we randomly select twenty non-overlapping $10 \times 10$ square regions as input locations. The output regions are then translated by $(h, v)$, where $h, v \in [-10, 10]$ denote horizontal and vertical shifts. Regions crossing image boundaries are mirrored to maintain continuity.

### 4.1.3. Real data 1: AirDelhi

The AirDelhi (Chauhan et al., 2024) dataset offers a comprehensive collection of fine-grained spatiotemporal particulate matter (PM) measurements from Delhi, India. To address the limitations of static sensor networks, the researchers mounted lower-cost PM2.5 sensors on public buses throughout the Delhi-NCR region. The dataset includes PM2.5 measurements across various locations and times, with data collected at a granularity of 20 samples per minute. The dynamic nature of the data, with sensors moving along predefined bus routes, introduces challenges such as sparse and temporally varying measurements. This necessitates the development of models capable of handling sparse and dynamically moving sensors. Here we use three data variants for model performance comparisons.

• **AirDelhi Benchmark (AD-B)**: This dataset was originally introduced as a benchmark for evaluation. Concretely, this data is collected between November 12, 2020, and January 30, 2021. In this data, initial days are omitted due to limited sample data and fewer instruments on buses. Also excluded are nightly data between 10:00 PM and 5:30 AM, as buses remain stationary at bus depots during this period. The geo-

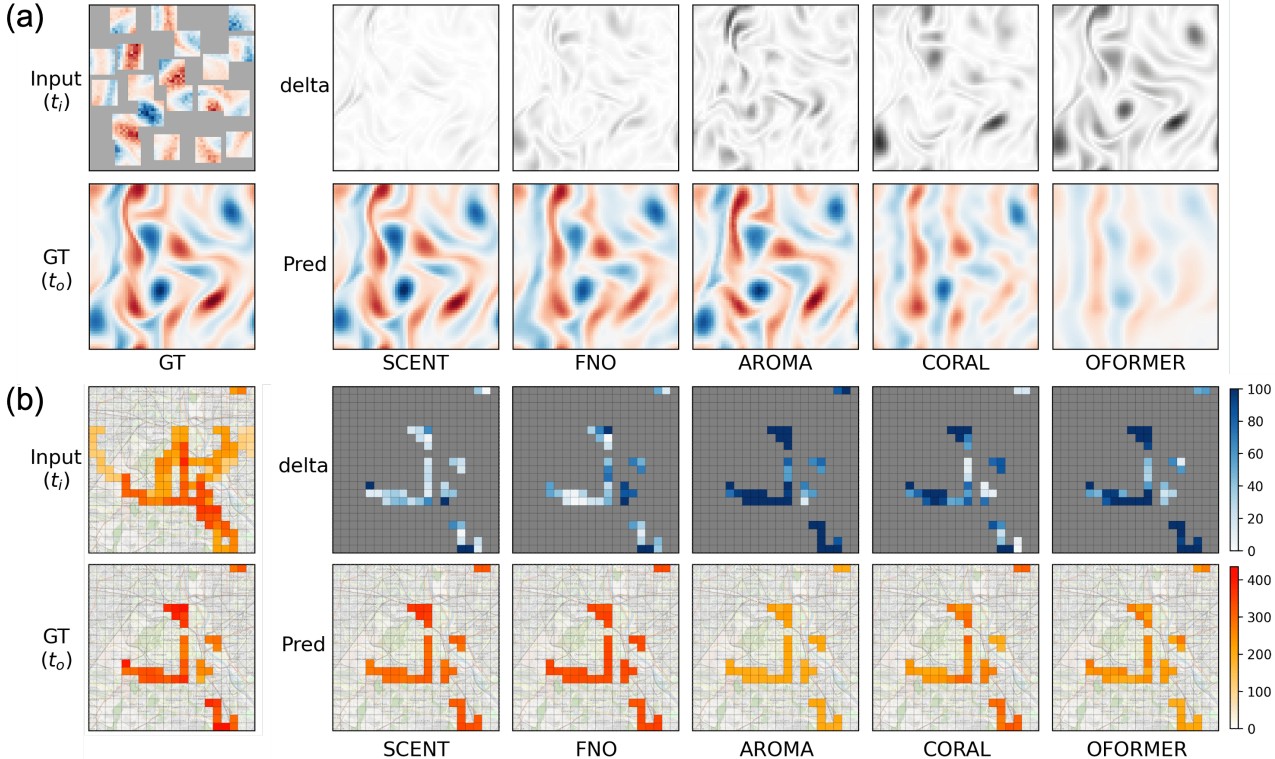

*Figure 5.* **Qualitative comparisons for forecasting.** (a) Using partial input from the S5 dataset, each pretrained model is assessed on the full mesh (GT), performing both joint forecasting and spatial interpolation. (b) Models forecast PM2.5 on the spatiotemporal locations where measurements are available (GT). 'delta' (upper rows) represents the absolute difference between each prediction and GT.

graphical region is divided into $1 \, \text{km}^2$ spatial grids, which are further segmented into spatiotemporal cells with a 30-minute time interval. The average of all samples within each spatiotemporal cell is calculated to determine representative PM values. We use the 'AB' and 'CP' sets as training and test datasets, respectively. This dataset features differing numbers of samples for input ($N_i$) and output ($N_o$), which demands a high degree of model flexibility. Notably, SCENT stands out as the only model capable of naturally handling a variable $N_i$ and $N_o$. • **AirDelhi Temporal (AD-T)**: this dataset still uses the $1 \, \text{km}^2$ grid, but increases time resolution from 30 minutes to a finer 1 minute, and averaging all samples within each spatiotemporal cell for representative PM values. This increases available data instances but makes data spatially more sparse and thus challenging to predict. Unlike the original dataset, where the train-test split was based on specific days, we randomly shuffle all the data before splitting it into training and testing sets. This approach ensures there is no distribution shift between the training and test data. • **AirDelhi Fine-grained (AD-F)**: this dataset features a $0.02 \, \text{km}^2$ spatial grid with a 1-minute temporal resolution. The high granularity of the spatial resolution makes it nearly continuous, enabling the evaluation of models in learning continuous representations effectively.

### 4.1.4. REAL DATA 2: KUROSHIO PATH

The Kuroshio current, originating from the North Equatorial Current (NEC) and flowing northward along the eastern side of the Philippine Islands, is the world's second-largest warm current. Accurately predicting its path is crucial because its variations significantly affect the exchange of water masses and heat between the North Pacific subtropical and subarctic circulations. We use 50-year records from the China Ocean Reanalysis (CORA) (Han et al., 2013) as our benchmark and follow the data processing guidelines established by Wu et al. (2023). CORA provides daily oceanographic reanalysis data for the Kuroshio current spanning 50 years (January 1958-December 2007).

### 4.1.5. REAL DATA 3: RAINFOREST NOWCASTING

We use the RY product from the German Weather Service (DWD), a quality-controlled rainfall composite at 1 km × 1 km spatial and 5-minute temporal resolution. Data from 2012 - 2016 are used for training, and 2017 for testing. The task is to predict rainfall fields for future timestamps $t \in [5, 10, \ldots, 60]$ minutes, given four historical fields. Following RainNet (Ayzel et al., 2020), we use 173,345 / 43,456 instances for training / test splits. We downsample the original 900 × 900 resolution to 64 × 64.

## 4.2. Robustness to Data Challenges

A robust model should reconstruct continuous fields from sparse, noisy data while capturing temporal dynamics. To evaluate this capability, we assess forecasting performance across five challenging datasets (S1-S5, Fig. 1(a)) using four representative baseline models. All models are trained with supervision on next-state prediction. Since FNO lacks mesh independence, we modify the original architecture to accommodate our sparse and irregular dataset. Sparse data variants are zero-padded to align with a regular grid representation for input processing. To prevent the model from trivially predicting zero values, a data mask is applied to the output during loss computation. The *time horizon* ($t_h$) is set to 3 for training. This implies that during training, the time step ($\Delta t$) is randomly selected from $[0, 3]$, allowing the model to learn time-continuity given varying temporal intervals as well as reconstruction ($\Delta t = 0$). Each training trajectory consists of 24 time steps, and a total of 100k trajectories are used. Training is conducted using Rel-MSE as the supervision loss, a cosine learning rate scheduler over 50k iterations, and a batch size of 256. During validation, $\Delta t$ is fixed to 1.

**Results.** Table 1 presents forecasting performance comparisons. SCENT consistently outperforms all baseline models on the simulated datasets, showcasing its resilience in challenging environments and its ability to effectively capture spatiotemporal patterns. As expected, performance generally degrades in more challenging environments, reflecting the increased difficulty of learning from sparse or noisy data. Surprisingly, FNO performs competitively despite its lack of mesh independence, likely due to its inherent bias toward learning low-frequency components, allowing it to maintain relatively strong performance.

## 4.3. Scalability Analysis

Scientific data volumes are rapidly increasing (Yu et al., 2023; Kitamoto et al., 2023; Kaltenborn et al., 2023), reaching the exabyte scale in cases such as ATLAS (Peters & Janyst, 2011). To assess scalability, we analyze model and dataset size variations, examining both model width (i.e., latent dimension) and depth (i.e., number of layers). To our knowledge, this is the first systematic scalability study of continuity-informed architectures (e.g., INRs, CNFs), and we compare against FNO by varying model width to assess large-scale adaptability. After constructing SCENT models of varying sizes (Appendix F), we design corresponding FNO models with approximately matching parameter counts for fair comparison.

**Results.** Fig. 4(a) shows both FNO and SCENT exhibit decreasing Rel-MSE as model size increases. However, SCENT consistently outperforms FNO across all model

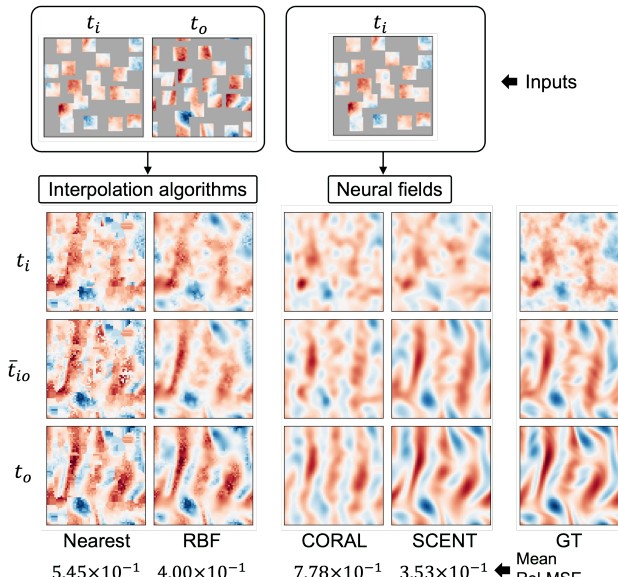

*Figure 6.* **Joint reconstruction, interpolation, and forecasting.** Given dataset S5 inputs shown on top, neural fields are tested for reconstruction/interpolation ($t_i$) and forecasting at continuous time ($\bar{t}_{io}, t_o$). For comparison purposes, interpolation results from nearest-neighborhood and RBF algorithms are shown on the left. 'Mean Rel-MSE' denotes average performance at $\bar{t}_{io}$ evaluated with the validation dataset.

sizes and shows better linear scalability, whereas FNO displays a clear convergence pattern. Additionally, Fig. 4(b) indicates larger datasets improve forecasting performance, as the model effectively leverages additional samples to capture complex underlying patterns from sparse sensor data.

## 4.4. Learning Spatiotemporal Continuity

Our primary objective is to represent a spatiotemporal field conditioned on a sparse input, as outlined in Section 2. An ideal model should, given an input at $t_i$, be capable of both reconstruction and spatial interpolation at $t_o$, as well as forecasting at any continuous time within the given time horizon. To evaluate this capability, we assess and compare model performance, with a particular focus on learning time-continuous representations. Specifically, given an S5 data input $U^{t_i}$, we infer the data field at three time points, $t_i, \bar{t}_{io}, t_o$, where $\bar{t}_{io} = (t_i + t_o)/2$ denotes the midpoint time. Reconstruction performances at $\bar{t}_{io}$ are examined against ground truth. We downsample each trajectory by a factor of two, reserving the rest for evaluating time continuity in learning. We compare the results against CORAL, which also learns time-continuous conditioned neural fields (Table 4). Additionally, we evaluate our model against deterministic interpolation methods, namely nearest-neighbor and radial basis function (RBF) interpolation. Unlike our approach, these methods require data at both $t_i$ and $t_o$ to interpolate the spatiotemporal full mesh effectively.

*Table 1.* Forecasting performance comparisons on simulated (Rel-MSE) and real datasets (RMSE).

| MODEL | SIMULATED CHALLENGING ENVIRONMENTS | | | | | REAL DATA (AIRDELHI) | | |
|---|---|---|---|---|---|---|---|---|
| | S1 | S2 | S3 | S4 | S5 | AD-B | AD-T | AD-F |
| FNO | $4.27 \times 10^{-2}$ | $2.10 \times 10^{-1}$ | $2.81 \times 10^{-1}$ | $2.35 \times 10^{-1}$ | $3.77 \times 10^{-1}$ | 48.79 | 55.04 | 55.66 |
| OFORMER | $1.63 \times 10^{-1}$ | $2.57 \times 10^{-1}$ | $3.30 \times 10^{-1}$ | $3.00 \times 10^{-1}$ | $6.17 \times 10^{-1}$ | 70.62 | 57.00 | 54.58 |
| CORAL | $4.12 \times 10^{-1}$ | $4.83 \times 10^{-1}$ | $4.69 \times 10^{-1}$ | $4.89 \times 10^{-1}$ | $9.06 \times 10^{-1}$ | 60.51 | 55.26 | 48.41 |
| AROMA | $1.29 \times 10^{-1}$ | $2.38 \times 10^{-1}$ | $2.83 \times 10^{-1}$ | $6.67 \times 10^{-1}$ | $5.25 \times 10^{-1}$ | **40.78** | 63.06 | 47.49 |
| SCENT | $\mathbf{2.51 \times 10^{-2}}$ | $\mathbf{2.08 \times 10^{-1}}$ | $\mathbf{2.70 \times 10^{-1}}$ | $\mathbf{2.28 \times 10^{-1}}$ | $\mathbf{3.26 \times 10^{-1}}$ | 44.20 | **53.24** | **45.35** |

*Table 2.* Long-term forecasting performances on Navier-Stokes benchmark datasets. MSE is used for NS-3 (for fair comparisons against reported prior arts), while Rel-MSE is used for NS-4 and NS-5.

| MODEL | BENCHMARK DATASETS | | |
|---|---|---|---|
| | NS-3 | NS-4 | NS-5 |
| FNO | $1.55 \times 10^{-4}$ | $1.53 \times 10^{-1}$ | $\underline{1.24 \times 10^{-1}}$ |
| DINO | $2.51 \times 10^{-2}$ | $7.25 \times 10^{-1}$ | $3.72 \times 10^{-1}$ |
| CORAL | $5.76 \times 10^{-4}$ | $3.77 \times 10^{-1}$ | $3.11 \times 10^{-1}$ |
| OFORMER | $7.76 \times 10^{-3}$ | $1.36 \times 10^{-1}$ | $2.40 \times 10^{-1}$ |
| GNOT | $3.21 \times 10^{-4}$ | $1.85 \times 10^{-1}$ | $1.65 \times 10^{-1}$ |
| AROMA | $\underline{1.32 \times 10^{-4}}$ | $\underline{1.05 \times 10^{-1}}$ | $1.24 \times 10^{-1}$ |
| SCENT | $\mathbf{7.78 \times 10^{-5}}$ | $\mathbf{1.03 \times 10^{-1}}$ | $\mathbf{1.17 \times 10^{-1}}$ |

*Table 3.* Ablation study for SCENT on dataset S5. ✓indicates activated modules. Abbreviations: CEN=Context Encoding Network; CN=Calibration Network; Proj=linear projection; TT=Time-Targeted: whether to provide $t_{\text{out}}$ for the TTSE.

| CEN | CN | PROJ | TT | REL-MSE | CONTRAST |
|---|---|---|---|---|---|
| ✓ | ✓ | ✓ | ✓ | $\mathbf{3.26 \times 10^{-1}}$ | - |
| ✗ | ✓ | ✓ | ✓ | $4.02 \times 10^{-1}$ | +23.3% |
| ✓ | ✗ | ✓ | ✓ | $3.64 \times 10^{-1}$ | +11.0% |
| ✓ | ✓ | ✗ | ✓ | $3.42 \times 10^{-1}$ | +4.9% |
| ✓ | ✓ | ✓ | ✗ | $\underline{3.40 \times 10^{-1}}$ | +4.29% |
| ✗ | ✗ | ✓ | ✓ | $4.61 \times 10^{-1}$ | +41.4% |
| ✗ | ✗ | ✗ | ✓ | $4.82 \times 10^{-1}$ | +47.8% |
| ✗ | ✗ | ✗ | ✗ | $5.47 \times 10^{-1}$ | +67.8% |

**Results.** Fig. 6 presents the reconstructed results for a given input. Notably, SCENT accurately interpolates and reconstructs all three time points, closely matching the ground truth (GT). While CORAL supports time interpolation, it inherits suboptimal performance from training with the complex S5 dataset (Table 1), resulting in an inaccurate reconstruction of fields across different time points. Interpolation using radial basis functions (RBF) yields reasonable results; however, it is important to note that the inputs for interpolation methods and neural fields differ. Neural fields operate at a disadvantage as they are provided only with the initial time state. To further evaluate time continuity, we report the mean reconstruction error against the GT at $\bar{t}_{io}$ on the full test dataset, with the quantitative results aligning well with the visual assessments.

## 4.5. Benchmark Performance and Model Ablations

In this section, we highlight the key innovations in our algorithm, compare it against popular benchmark datasets, and present ablation studies on the SCENT architecture. To evaluate SCENT's ability to perform extended time forecasting, we test it on NS-3,4,5 (Section 4.1.1) using an unrolling approach. Specifically, all models are trained with supervision on next-state prediction. At test time, we unroll the dynamics following the WUF framework, as illustrated in Section 3.3. Additionally, we conduct an ablation study on S5, systematically removing our key architectural components — CEN, CN, Proj, and TT (Fig. 2) — to assess their individual contributions to performance.

**Results.** Table 2 presents the results of long-term forecasting performance across the benchmark datasets. SCENT outperforms all baseline models across all datasets, which we attribute to WUF, fundamentally enabled by the time-continuity learned by the model. This advantage is particularly evident in the NS-3 dataset, where the fluid dynamics are relatively slower, hence error accumulation during next-state unrolling is more pronounced. On NS-4 and NS-5, SCENT and AROMA achieve comparable performances. Our ablation study in Table 3 highlights the contributions of individual architectural modifications introduced in SCENT. The performance gap relative to the best-performing model, referred to as *contrast*, demonstrates that all four components play a crucial role in the model's effectiveness. Notably, performance deteriorates significantly when two or more modules are deactivated, with Rel-MSE increasing by 67.8% when all modules are not used.

## 4.6. Forecasting Performances on AirDelhi

We evaluate whether the superior performance observed in previous experiments extends to the more complex AirDelhi dataset. Similar to dataset S5, AirDelhi features sparse sensors with locations that vary across time, posing a significant challenge for spatiotemporal learning. Strong performance on this dataset would indicate that the model effectively infers the PM2.5 distribution from sparse observations and accurately predicts the future diffusion of particulate matter. While we use Mean Squared Error (MSE) as the training

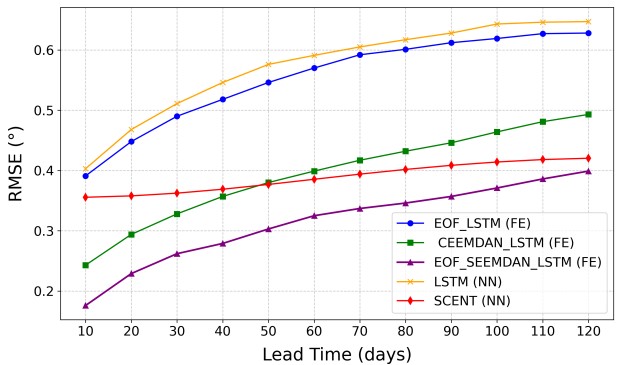

*Figure 7.* Kuroshio Path: RMSE (°) comparisons at various lead times. Abbreviations: FE = Feature engineering, NN = End-to-end neural networks, EOF = empirical orthogonal functions, CEEM-DAN = complete ensemble empirical mode decomposition with adaptive noise, LSTM = long short-term memory.

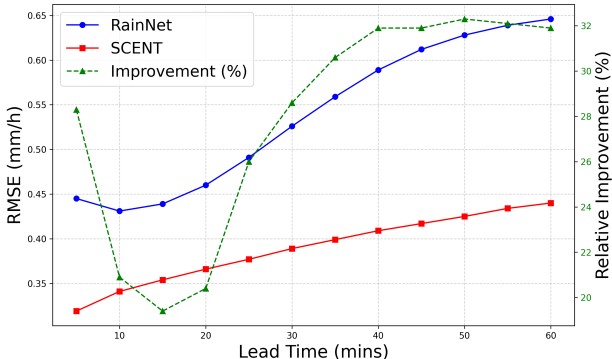

*Figure 8.* Rainforest nowcasting: comparing RainNet and SCENT performances at varying lead times. Green line corresponds to the relative gain of SCENT against RainNet baseline measured in percentage improvement.

loss, we adopt Root Mean Squared Error (RMSE) for evaluation to better capture the physical significance of the model's performance.

**Results.** Table 1 compares forecasting performance across AirDelhi datasets. On AD-B, where the goal is to predict PM2.5 values at sparse locations given three days of observations, SCENT ranks second after AROMA, likely due to AROMA's diffusion backbone effectively filtering noise. However, the qualitative results in Fig. 5(b) highlight SCENT's effectiveness in capturing and expressing PM2.5 levels. All of the five models outperform previously reported performances on AD-B (Appendix K). For larger and finer datasets (AD-T and AD-F), SCENT outperforms all baselines, demonstrating its strength in learning continuous representations and forecasting. Additional results (Appendix Fig. J) indicates that SCENT better captures the PM2.5 distribution.

### 4.7. Longterm Forecasting on the Kuroshio Path

We compared SCENT against four baseline methods from Wu et al. (2023) in a 120-day Kuroshio path prediction experiment, training on 40 years of data (1958-1997) and testing on the subsequent 10 years (1998-2007). We measured RMSE against true latitude at 10-day intervals. During training, SCENT is used to predict the Kuroshio path in terms of latitude (ranging from 29°N to 36°N) for a fixed forecast horizon. As shown in Fig. 8, while CEEMDAN-based feature engineering (FE) variants performed competitively, SCENT achieved second-best performance beyond a 50-day lead time. The FE methods' strength might stem from the limited dataset size, but our scalability study (Fig. 4) suggests SCENT could outperform them with more data. Importantly, SCENT maintained stable performance even as the lead time increased, a sharp contrast to other

baselines experiencing significant degradation.

### 4.8. Rainforest Nowcasting

SCENT is trained with a forecast horizon $t_h = 60$. We report RMSE (mm h$^{-1}$) in Figure 8. SCENT consistently outperforms RainNet across all lead times, with the relative improvement more salient with a larger lead time. We attribute this gain in part to SCENT's ability to train with variable target times $t_o$, which serves as a form of data augmentation.

## 5. Conclusion

SCENT (Scalable Conditioned Neural Field for Spatiotemporal Learning) addresses the challenge of reconstructing and forecasting spatiotemporal fields from sparse and noisy data. Through extensive evaluations, we demonstrate SCENT's superiority in learning continuous space-time representations, outperforming baselines in diverse forecasting and reconstruction tasks. SCENT is the first single-step training model for learning continuous spatiotemporal representations, eliminating multi-stage optimization bottlenecks. Its scalability makes it suitable for large-scale applications in geophysics, astronomy, epidemiology, and nuclear physics (Reichstein et al., 2019; Gabbard et al., 2022; Massucci et al., 2016; Pata et al., 2024). Future work will focus on expanding SCENT's adaptability to extreme-scale datasets and real-world deployments.

Further research will also explore enhancing SCENT's capabilities for high-frequency phenomena, often challenging for neural fields. We will investigate integrating multimodal data sources, allowing SCENT to leverage diverse information for robust, accurate spatiotemporal modeling. We will also tackle multi-scale training problems, enabling SCENT to capture both fine-grained details and large-scale trends within complex spatiotemporal systems.

## Acknowledgements

This work was supported by the U.S. Department of Energy (DOE), Office of Science (SC), Advanced Scientific Computing Research program under award DE-SC-0012704 and used resources of the National Energy Research Scientific Computing Center, a DOE Office of Science User Facility using NERSC award NERSC DDR-ERCAP0030592.

## Impact Statement

This paper introduces a robust, scalable, and unified machine learning approach for understanding and predicting complex spatiotemporal dynamics found in critical scientific datasets. Such advancements hold the potential to enhance predictive insights into complex systems and advance fundamental scientific knowledge.

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

# A. Related Work

## A.1. Spatiotemporal Learning

Spatiotemporal learning has emerged as a fundamental area of research, addressing the need to model and interpret data that evolve across both space and time. This field has far-reaching applications in diverse scientific and engineering disciplines, enabling advancements in climate modeling (Reichstein et al., 2019), traffic forecasting (Li et al., 2018), medical imaging (Litjens et al., 2017), and video analysis (Wang et al., 2020). Beyond these areas, spatiotemporal learning plays a crucial role in biochemistry, where modeling molecular interactions and protein dynamics requires capturing complex temporal dependencies in high-dimensional spatial structures (Jumper et al., 2021). In nuclear particle physics, tracking subatomic particles in high-energy collisions demands precise spatiotemporal reconstruction to infer particle trajectories and decay chains (Aad et al., 2019). Neuroscience increasingly relies on spatiotemporal models to analyze large-scale neural recordings, such as EEG and fMRI, for understanding brain activity over time and across different brain regions (Van Essen et al., 2010). Similarly, in epidemiology, disease spread modeling depends on accurately capturing transmission dynamics across spatially distributed populations over time (Balcan et al., 2009).

Other critical applications include remote sensing and Earth observation, where satellite imagery and geospatial data must be processed to track environmental changes, deforestation, and urbanization trends (Woodcock et al., 2008). In fluid dynamics, understanding turbulent flow patterns and their evolution over time is key to designing efficient aerodynamic structures and predicting oceanic and atmospheric circulation (Meneveau & Marusic, 2011). These applications highlight the growing importance of spatiotemporal learning in scientific discovery and engineering innovation. However, effectively capturing the complex dependencies inherent in spatiotemporal data presents significant challenges, including spatial heterogeneity (irregularly structured and non-uniformly distributed data), temporal non-stationarity (changes in statistical properties over time), and high dimensionality (large-scale data with intricate interdependencies). Addressing these challenges requires scalable, generalizable, and computationally efficient modeling approaches that can learn from noisy, sparse, and dynamically evolving spatiotemporal datasets.

## A.2. Traditional Approaches to Spatiotemporal Learning

Historically, Recurrent Neural Networks (RNNs) and Long Short-Term Memory networks (LSTMs) have been employed to model temporal sequences (Hochreiter & Schmidhuber, 1997). For instance, LSTMs have been widely used in traffic forecasting to predict future traffic conditions based on historical data (Yu et al., 2017). Despite their effectiveness, these models often struggle with capturing long-range dependencies and may not fully exploit spatial correlations. To address these limitations, Transformer-based architectures have been introduced, leveraging self-attention mechanisms to model long-range dependencies in both spatial and temporal dimensions. Vision Transformer (ViT) has demonstrated success in image analysis by treating images as sequences of patches, enabling the modeling of global relationships (Dosovitskiy et al., 2021). Extending this idea, TimeSformer has been proposed for video understanding, jointly modeling spatial and temporal dependencies to achieve state-of-the-art results in action recognition tasks (Bertasius et al., 2021).

## A.3. Implicit Neural Representations (INRs) for Learning Continuous Representations

Implicit Neural Representations (INRs) have emerged as a flexible and powerful approach for modeling continuous signals. Unlike traditional grid-based representations, INRs parameterize data as continuous functions, mapping spatial coordinates to function values using a neural network (Sitzmann et al., 2020b). This formulation allows for compact data encoding, resolution-agnostic modeling, and seamless interpolation. INRs have been widely applied in 3D shape representation, scene reconstruction, and signal processing, where high-dimensional structured data needs to be represented efficiently. A notable example is Neural Radiance Fields (NeRF), which employs INRs for view synthesis, enabling the rendering of high-fidelity 3D scenes from sparse observations (Mildenhall et al., 2020).

Despite these advantages, INRs face challenges in generalizability, as a trained INR typically encodes only a single instance of data and does not naturally adapt to new instances. This limitation necessitates retraining the model for each new data sample, making INRs computationally expensive for large-scale applications. Generalizable INRs (GINRs) aim to overcome this by introducing mechanisms that allow a single model to adapt across multiple instances, rather than learning a fixed function for each data sample. Locality-Aware Generalizable Implicit Neural Representations introduce local feature conditioning that allows GINRs to adapt dynamically to different regions of a dataset, improving efficiency and generalization (Lee et al., 2023). Furthermore, MetaSDF and MetaSIREN employ gradient-based meta-learning to

enable few-shot adaptation, allowing GINRs to learn priors across multiple data instances and generalize to unseen samples more effectively (Sitzmann et al., 2020a; Chen et al., 2021). Functa treats each data instance as a function and leveraging function-space representations for improved generalization (Dupont et al., 2022). Spatial Functa extends this approach to large-scale datasets like ImageNet, introducing spatially aware latent spaces that enhance expressivity and enable GINRs to perform large-scale classification and generation tasks (Bauer et al., 2023).

### A.4. Conditioned Neural Fields as GINRs

Recent work by Wang et al. (2024)(Wang et al., 2024) highlights the close relationship between CNFs and INRs, emphasizing that both frameworks model continuous fields but differ in their conditioning mechanisms. While INRs typically encode static signals without external conditioning, CNFs introduce input-dependent variations, making them well-suited for physics-informed learning in PDE solving. This connection unifies perspectives on operator learning and neural field-based modeling, bridging the gap between classical numerical methods and neural representations.

Conditioned Neural Fields (CNFs) provide a powerful framework for solving Partial Differential Equations (PDEs) by learning continuous function representations conditioned on input parameters, initial conditions, or constraints (Li et al., 2022). Unlike traditional numerical solvers that rely on discretization, CNFs approximate time-evolving fields in a resolution-independent manner, making them particularly effective for high-dimensional and sparse-data problems. This approach aligns closely with GINRs, which parameterize data as continuous functions rather than grid-based representations (Sitzmann et al., 2020b).

### A.5. Baseline Selection

Spatiotemporal learning requires modeling dynamic systems such as fluid flows, climate forecasting, and wave propagation, all of which are governed by PDEs. PDE-based neural solvers provide strong priors that enhance generalization, consistency, and interpretability. To evaluate our approach (SCENT) against established methods, we compare against FNO, which learns spectral representations for PDE solutions, OFormer, a Transformer-based operator learner, and CORAL and AROMA, which leverage neural fields for dynamic system modeling. These baselines offer diverse perspectives on how different architectures generalize across spatiotemporal interpolation, reconstruction, and forecasting tasks.

*Table 4.* Comparing model capacities for learning spatiotemporal continuity from discrete data.

| MODEL | MESH AGNOSTIC LEARNING | SPACE&TIME CONTINUOUS LEARNING | SINGLE-STEP TRAINING |
|---|---|---|---|
| FNO (LI ET AL., 2020) | ✗ | ✗ | ✓ |
| OFORMER (LI ET AL., 2023) | ✓ | ✗ | ✓ |
| CORAL (SERRANO ET AL., 2023) | ✓ | ✓ | ✗ |
| AROMA (SERRANO ET AL., 2024) | ✓ | ✗ | ✗ |
| SCENT (OURS) | ✓ | ✓ | ✓ |

# B. SCENT: Pseudo-Algorithm

We omit spatiotemporal mesh variables $\left( \{\mathbf{x}_j\}_{j=1}^{N_i}, t_i, t_o \right)$ in the algorithm for simplification.

---

**Algorithm 1** Training and Inference for Model $F$

---

**Training Procedure:**

Model $F$, training dataset $\mathcal{D}$ with $n_{tr}$ trajectories, each of $T$ timesteps

Time horizon $t_h$, total training steps $N_{train}$, loss function $g$

Initialize model parameters $\theta$

**for** iteration $i = 1$ to $N_{train}$ **do**

    Select $i$th trajectory from $\mathcal{D}$

    Sample $(U^{t_i}, U^{t_o})$ with time horizon $t_h$, where $t_o - t_i = \Delta t \in \{0, 1, 2, \ldots, t_h\}$

    **Forward Pass:** Compute model prediction $\hat{U}^{t_o} = F(U^{t_i})$

    Compute loss: $\mathcal{L} = g(\hat{U}^{t_o}, U^{t_o})$

    **Backpropagation:** Compute gradients $\nabla_\theta \mathcal{L}$ and update $\theta$

**end for**

**return** Trained model $F$

**Validation Procedure:**

Validation dataset $\mathcal{V}$ with $n_{val}$ trajectories with timesteps T, trained model $F$, validation metric $g$

Freeze model $F$ {Disable gradient updates}

Initialize empty list $B = [\,]$

**for** each validation trajectory in $\mathcal{V}$ **do**

    **for** each $(U^{t_i}, U^{t_o})$ where $t_i \in [0, T-1]$ and $t_o = t_i + 1$ **do**

        Forward propagate: $\hat{U}^{t_o} = F(U^{t_i})$

        Compute loss: $\mathcal{L} = g(\hat{U}^{t_o}, U^{t_o})$

        Append $\mathcal{L}$ to list $B$

    **end for**

**end for**

**return** $\frac{1}{|B|} \sum_{b \in B} b$ {Return the average}

**Forecasting Procedure:**

Trained model $F$, a set of evaluation samples $\mathcal{E}$ with desired targeted time $t_o$

Initialize empty list $B = [\,]$

**for** each evaluation sample $U^{t_i}$ in $\mathcal{E}$ **do**

    Compute quotient and remainder: $q = \lfloor (t_o - t_i)/t_h \rfloor$, $t_r = (t_o - t_i) \mod t_h$

    **for** $j = 1$ to $q$ **do** {Performing Warp-Unrolling Forecasting (Section 3.3)}

        Update $t_c \leftarrow t_i + t_h$

        Forward propagate: $\hat{U}^{t_c} = F(U^{t_i})$

        Update $t_i \leftarrow t_c$

    **end for**

    **if** $r > 0$ **then**

        Forward propagate: $\hat{U}^{t_o} = F(U^{t_i + t_r})$ {One last forward pass with the remainder $t_r$}

    **end if**

    Append $\hat{U}^{t_o}$ to list $B$

**end for**

**return** list $B$ {Return predicted samples}

---

## C. Data statistics and Hyperparameters for training SCENT on Navier-Stokes Benchmark Datasets

|  |  | DATASET NAME | | |
| --- | --- | --- | --- | --- |
|  |  | NS-3 | NS-4 | NS-5 |
| DATA STATISTICS | # TRAJECTORIES - TRAIN | 256 | 1000 | 1000 |
|  | # TRAJECTORIES -VALIDATION | 64 | 200 | 200 |
|  | MAXIMUM T | 30 | 30 | 20 |
|  | INITIAL T | 10 | 10 | 10 |
|  | SPATIAL RESOLUTION | (64,64) | (64,64) | (64,64) |
|  | N POINTS - INPUTS (N) | 4096 | 4096 | 4096 |
|  | N POINTS - INPUTS (M) | 4096 | 4096 | 4096 |
| TRAINING | MAX LR | 0.001 | 0.0006 | 0.0008 |
|  | MIN LR | 0 | 0 | 0.000008 |
|  | LR SCHEDULER | COSINE | COSINE | COSINE |
|  | WARMUP STEPS | 0 | 2000 | 2000 |
|  | BATCH SIZE | 100 | 100 | 100 |
|  | TOTAL STEPS | 150000 | 110000 | 110000 |
|  | OPTIMIZER | ADAMW | ADAMW | ADAMW |
|  | BETA1 | 0.9 | 0.9 | 0.9 |
|  | BETA2 | 0.999 | 0.999 | 0.999 |
|  | TRAINING TIME HORIZON | 5 | 5 | 5 |
|  | WEIGHT DECAY | 0.00001 | 0.00001 | 0.00001 |
| MODEL | EMBED DIM | 128 | 128 | 128 |
|  | LATENT DIM | 128 | 128 | 128 |
|  | LINEAR PROJECTION DIM | 64 | 64 | 64 |
|  | # LEARNABLE QUERIES | 64 | 256 | 256 |
|  | # LAYERS - PROCESSOR | 2 | 2 | 2 |
|  | # LAYERS - ENCODER | 4 | 4 | 4 |
|  | # LAYERS - DECODER | 4 | 4 | 4 |
|  | # HEADS | 4 | 4 | 4 |
|  | SPARSE ATTENTION - GROUP SIZE | 1 | 8 | 8 |
|  | FF MULTIPLIER | 4 | 4 | 4 |
| EMBEDDING | OUTPUT SCALE | 0.1 | 0.1 | 0.1 |
|  | LATENT INIT SCALING (STD) | 0.02 | 0.02 | 0.02 |
|  | FOURIER FEATURES # FREQUENCY BANDS | 6 | 12 | 12 |
|  | FOURIER FEATURES MAX RESOLUTION | 20 | 20 | 20 |

## D. Data statistics and Hyperparameters for Training SCENT on Simulated Datasets

| | | DATASET NAME | | | | |
|---|---|---|---|---|---|---|
| | | S1 | S2 | S3 | S4 | S5 |
| DATA STATISTICS | # TRAJECTORIES - TRAIN | 100000 | 100000 | 100000 | 100000 | 100000 |
| | # TRAJECTORIES -VALIDATION | 1000 | 1000 | 1000 | 1000 | 1000 |
| | MAXIMUM T | 50 | 50 | 50 | 50 | 50 |
| | INITIAL T | 1 | 1 | 1 | 1 | 1 |
| | TEMPORAL SUBSAMPLE STEP SIZE | 2 | 2 | 2 | 2 | 2 |
| | SPATIAL RESOLUTION | (64,64) | (64,64) | (64,64) | (64,64) | (64,64) |
| | N POINTS - INPUTS ($N$) | 4096 | 4096 | 2840 | 2048 | 2000 |
| | N POINTS - INPUTS ($M$) | 4096 | 4096 | 2840 | 2048 | 2000 |
| TRAINING | MAX LR | 0.0008 | 0.0008 | 0.0008 | 0.0008 | 0.0008 |
| | MIN LR | 0.00008 | 0.00008 | 0.00008 | 0.00008 | 0.00008 |
| | LR SCHEDULER | COSINE | COSINE | COSINE | COSINE | COSINE |
| | WARMUP STEPS | 2000 | 2000 | 2000 | 2000 | 2000 |
| | BATCH SIZE | 256 | 256 | 256 | 256 | 256 |
| | TOTAL STEPS | 100000 | 100000 | 50000 | 50000 | 50000 |
| | OPTIMIZER | ADAMW | ADAMW | ADAMW | ADAMW | ADAMW |
| | BETA1 | 0.9 | 0.9 | 0.9 | 0.9 | 0.9 |
| | BETA2 | 0.999 | 0.999 | 0.999 | 0.999 | 0.999 |
| | TRAINING TIME HORIZON | 3 | 3 | 3 | 3 | 3 |
| | WEIGHT DECAY | 0.0001 | 0.0001 | 0.0001 | 0.0001 | 0.0001 |
| MODEL | EMBED DIM | 164 | 164 | 164 | 164 | 164 |
| | LATENT DIM | 128 | 128 | 128 | 128 | 128 |
| | LINEAR PROJECTION DIM | 64 | 64 | 64 | 64 | 64 |
| | # LEARNABLE QUERIES | 128 | 128 | 128 | 128 | 128 |
| | # LAYERS - PROCESSOR | 2 | 2 | 2 | 2 | 2 |
| | # LAYERS - ENCODER | 6 | 6 | 6 | 6 | 6 |
| | # LAYERS - DECODER | 6 | 6 | 6 | 6 | 6 |
| | # HEADS | 4 | 4 | 4 | 4 | 4 |
| | SPARSE ATTENTION - GROUP SIZE | 2 | 2 | 8 | 8 | 8 |
| | FF MULTIPLIER | 4 | 4 | 4 | 4 | 4 |
| EMBEDDING | OUTPUT SCALE | 0.1 | 0.1 | 0.1 | 0.1 | 0.1 |
| | LATENT INIT SCALING (STD) | 0.02 | 0.02 | 0.02 | 0.02 | 0.02 |
| | FOURIER FEATURES # FREQUENCY BANDS | 12 | 12 | 12 | 12 | 12 |
| | FOURIER FEATURES MAX RESOLUTION | 20 | 20 | 20 | 20 | 20 |

## E. Additional Descriptions on Simulated Datasets

This dataset represents an incompressible fluid dynamics system governed by the vorticity transport equation:

$$\frac{\partial \omega}{\partial t} = -\mathbf{u} \cdot \nabla \omega + \nu \Delta \omega + f,$$

where the vorticity is defined as:

$$\omega = \nabla \times \mathbf{u}, \quad \nabla \cdot \mathbf{u} = 0. \qquad (2)$$

Here, $\mathbf{u}$ denotes the velocity field, and $\omega$ represents the vorticity. Both quantities are defined on a spatial domain with periodic boundary conditions. The parameter $\nu$ represents the kinematic viscosity, and $f$ is an external forcing function applied to sustain turbulence.

The input at time $t$ is given as $v_t = \omega_t$. The spatial domain is defined as:

$$\Omega = [-\pi, \pi]^2.$$

The vorticity field is initialized using a Gaussian Random Field (GRF) with parameters: $alpha = 2.5, \quad \tau = 3.0$.

Here, $\alpha$ controls the smoothness of the initial vorticity distribution, while $\tau$ determines the correlation length scale of the spatial structures. The external forcing function used in the simulation is:

$$f(x_1, x_2) = -4\cos(4x_2), \tag{3}$$

where $x_2$ represents the vertical spatial coordinate.

This forcing function introduces a structured periodic forcing along the vertical direction, promoting rotational flow characteristics. The periodic nature of the cosine function ensures a repeating vortex structure, which sustains turbulence and prevents energy dissipation over time. The negative sign maintains a consistent direction of vorticity input, reinforcing the rotational dynamics within the system. As a result, this setup generates a persistent and well-defined turbulent flow pattern. A Reynolds number of $Re = 100$ is used, indicating a moderately turbulent regime where inertial forces are dominant over viscous forces, allowing for complex vortex interactions while maintaining numerical stability. This is particularly relevant for spatiotemporal learning, as it provides a complex yet structured temporal evolution of the vorticity field, making it an ideal testbed for evaluating models that aim to learn continuous representations of dynamic physical systems. We generate in total 100k trajectories, with 50 time steps per trajectory. Among $T = 50$ data, we use every two steps ($\{0, 2, 4, \ldots, 48\}$) for training and validation. The remaining time steps ($\{1, 3, 5, \ldots, 49\}$) are reserved for evaluating the model's ability to learn continuous temporal representations.

## F. Hyperparameters for Model Scalability Evaluations(Fig. 4(a))

We use dataset S5 for the model scalability evaluations.

| | | MODEL NAME | | | | | | |
|---|---|---|---|---|---|---|---|---|
| | | M1 | M2 | M3 | M4 | M5 | M6 | M7 |
| TRAINING | MAX LR | 0.0008 | 0.0007 | 0.0006 | 0.0005 | 0.0004 | 0.0003 | 0.0002 |
| | MIN LR | 0.00008 | 0.00007 | 0.00006 | 0.00005 | 0.00004 | 0.00003 | 0.00002 |
| | LR SCHEDULER | COSINE | COSINE | COSINE | COSINE | COSINE | COSINE | COSINE |
| | WARMUP STEPS | 2000 | 2000 | 2000 | 2000 | 2000 | 2000 | 2000 |
| | BATCH SIZE | 256 | 256 | 256 | 256 | 256 | 256 | 256 |
| | TOTAL STEPS | 50000 | 50000 | 50000 | 50000 | 50000 | 50000 | 50000 |
| | OPTIMIZER | ADAMW | ADAMW | ADAMW | ADAMW | ADAMW | ADAMW | ADAMW |
| | BETA1 | 0.9 | 0.9 | 0.9 | 0.9 | 0.9 | 0.9 | 0.9 |
| | BETA2 | 0.999 | 0.999 | 0.999 | 0.999 | 0.999 | 0.999 | 0.999 |
| | TRAINING TIME HORIZON | 3 | 3 | 3 | 3 | 3 | 3 | 3 |
| | WEIGHT DECAY | 0.0001 | 0.0001 | 0.0001 | 0.0001 | 0.0001 | 0.0001 | 0.0001 |
| MODEL | LATENT DIM | 128 | 192 | 256 | 384 | 512 | 768 | 1024 |
| | # LEARNABLE QUERIES | 128 | 138 | 164 | 192 | 224 | 224 | 256 |
| | # LAYERS - PROCESSOR | 2 | 2 | 2 | 2 | 2 | 4 | 6 |
| | # LAYERS - ENCODER | 6 | 6 | 6 | 6 | 6 | 6 | 6 |
| | # LAYERS - DECODER | 6 | 6 | 6 | 6 | 6 | 6 | 6 |
| | # HEADS - PROCESSOR | 4 | 4 | 4 | 6 | 8 | 8 | 12 |
| | # HEADS - ENCODER | 2 | 2 | 2 | 2 | 4 | 4 | 4 |
| | # HEADS - DECODER | 2 | 2 | 2 | 2 | 4 | 4 | 4 |
| | EMBED DIM | 164 | 164 | 164 | 164 | 164 | 164 | 164 |
| | LINEAR- PROJECTION DIM | 64 | 64 | 64 | 64 | 64 | 64 | 64 |
| | SPARSE ATTENTION GROUP SIZE | 8 | 8 | 8 | 8 | 8 | 8 | 8 |
| | FF MULTIPLIER | 4 | 4 | 4 | 4 | 4 | 4 | 4 |
| EMBEDDING | OUTPUT SCALE | 0.1 | 0.1 | 0.1 | 0.1 | 0.1 | 0.1 | 0.1 |
| | LATENT INIT SCALING (STD) | 0.02 | 0.02 | 0.02 | 0.02 | 0.02 | 0.02 | 0.02 |
| | FOURIER FEATURES # FREQUENCY BANDS | 12 | 12 | 12 | 12 | 12 | 12 | 12 |
| | FOURIER FEATURES MAX RESOLUTION | 20 | 20 | 20 | 20 | 20 | 20 | 20 |

## G. Hyperparameters for Dataset Scalability Evaluations  (Fig. 4(b))

Other hyperparameters follow Appendix F for each corresponding model size.

| MODEL NAME | LATENT DIM | DATASET NAME | # TRAJECTORIES | BATCH SIZE | TOTAL STEPS | MAX LR | MIN LR |
|---|---|---|---|---|---|---|---|
| M1 | 128 | S5-30 | 30000 | 76 | 50000 | 0.0008 | 0.00008 |
| | | S5-50 | 50000 | 128 | 50000 | 0.0008 | 0.00008 |
| | | S5-100 | 100000 | 256 | 50000 | 0.0008 | 0.00008 |
| M2 | 256 | S5-30 | 30000 | 76 | 50000 | 0.0006 | 0.00006 |
| | | S5-50 | 50000 | 128 | 50000 | 0.0006 | 0.00006 |
| | | S5-100 | 100000 | 256 | 50000 | 0.0006 | 0.00006 |
| M3 | 512 | S5-30 | 30000 | 76 | 50000 | 0.0004 | 0.00004 |
| | | S5-50 | 50000 | 128 | 50000 | 0.0004 | 0.00004 |
| | | S5 - 100 | 100000 | 256 | 50000 | 0.0004 | 0.00004 |

# H. Learning Spatiotemporal Continuity: additional results

Two additional instances are illustrated. Please refer to main manuscript Fig. 6 for interpretation.

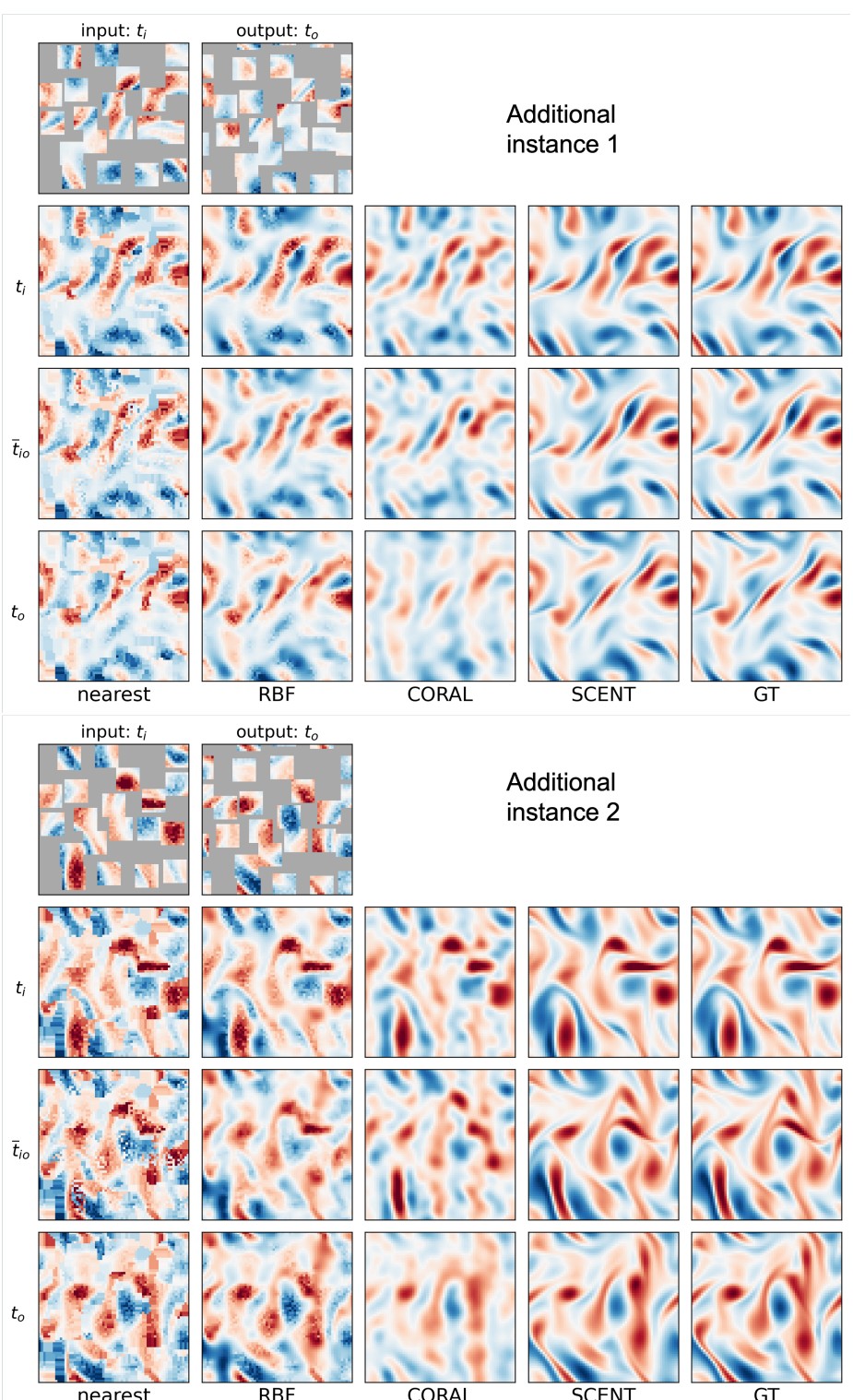

# I. Forecasting on S5 dataset: additional results

Four additional instances are illustrated. Please refer to main manuscript Fig. 5(a) for full details and interpretations.

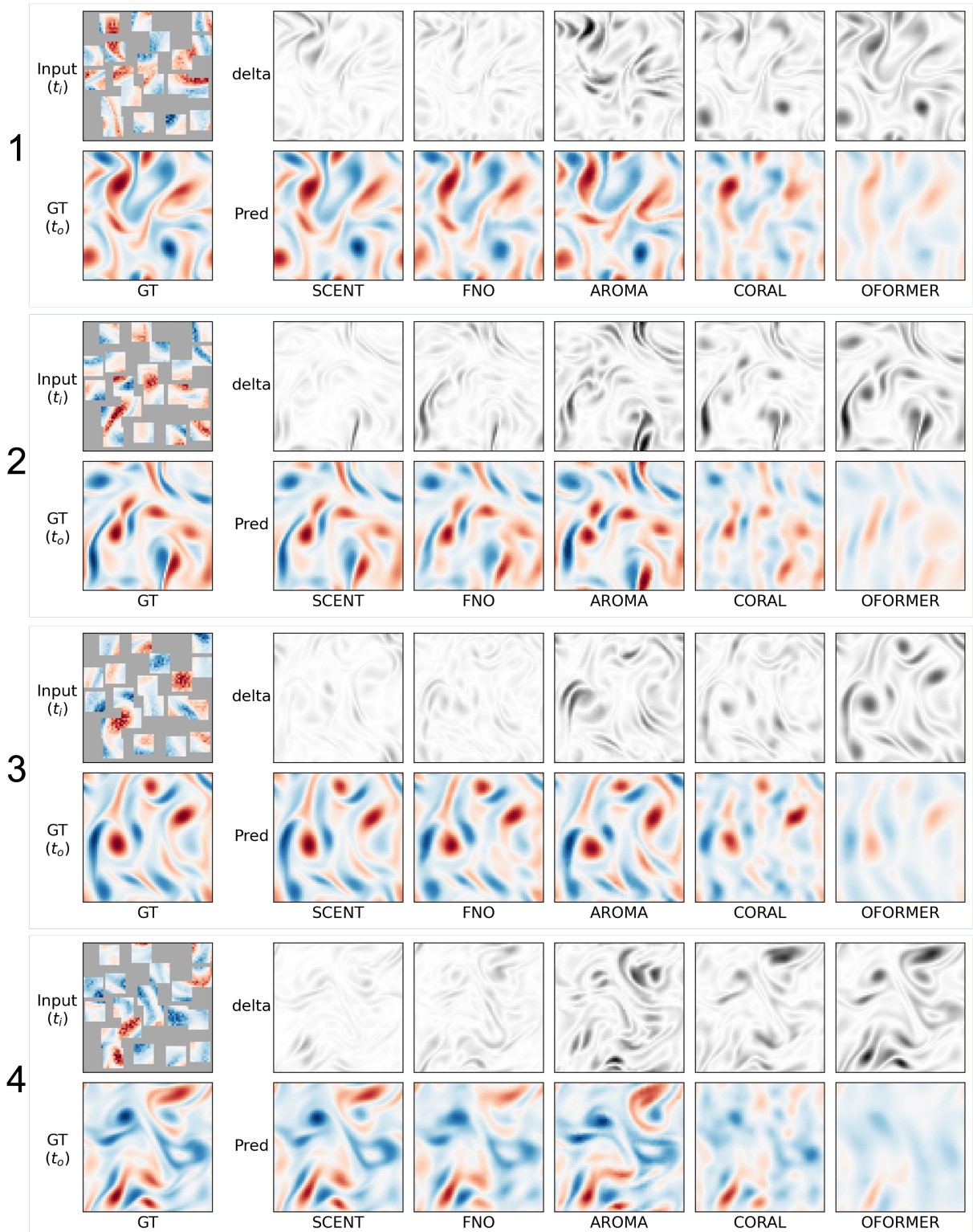

# J. Forecasting on AD-B dataset: additional results

Four additional instances are illustrated. Please refer to main manuscript Fig. 5(b) for details and interpretations.

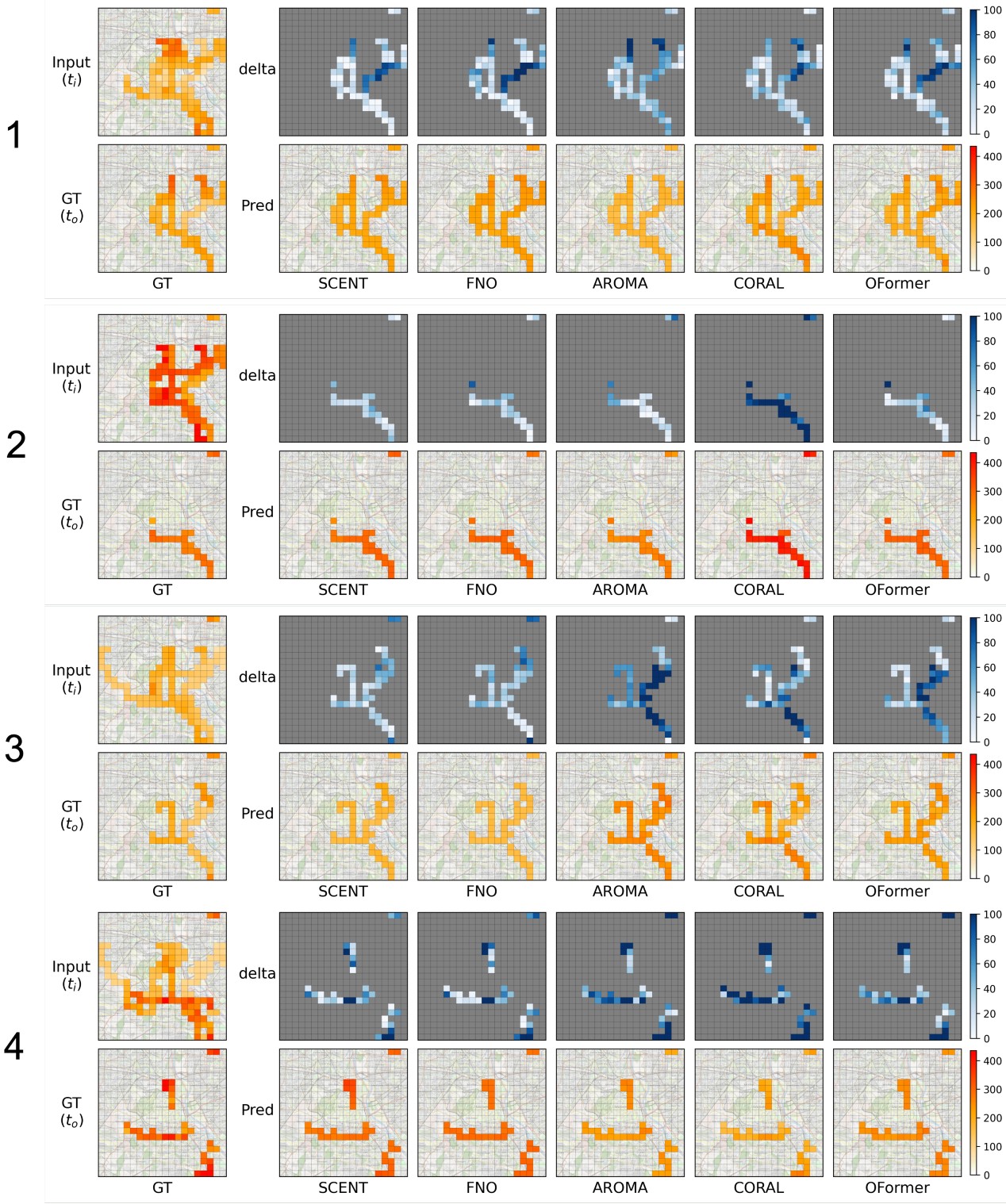

## K. AirDelhi AD-B: Comparisons Against Previously Reported Benchmark Performances

Here, we report and compare performances against SCENT and other baselines. Our experiments with conditioned neural fields set new record in AD-B benchmark. Inverse Distance Weighting (IDW) computes the weighted average of all visible samples based on their distances, assigning this value to the held-out locations. Random Forest (RF) is a non-linear model designed to capture complex spatial relationships. It excels in non-linear regression tasks by utilizing an ensemble of decision trees, with the final prediction obtained as the mean output across all trees. XGBoost (XGB) incrementally enhances predictions by combining weak estimators. During training, it employs gradient boosting to optimize performance while sequentially adding new trees. ARIMA (Auto-Regressive Integrated Moving Average) is a statistical model for time-series forecasting that applies linear regression. N-BEATS (Neural Basis Expansion Analysis for Time Series) (Oreshkin et al., 2020) is a deep learning model designed for zero-shot time-series forecasting. Non-Stationary Gaussian Process (NSGP) (Patel et al., 2022) is a recent Gaussian process baseline that models a non-stationary covariance for latitude and longitude, along with a locally periodic covariance for time.

| PRIOR REPORTED PERFORMANCES | | UPDATED PERFORMANCES | |
|---|---|---|---|
| MODEL | RMSE | MODEL | RMSE |
| IDW | 86.52 | FNO | 48.79 |
| RF | 110.49 | OFORMER | 70.62 |
| XGBOOST | 102.68 | CORAL | 60.51 |
| NSGP | 95.83 | AROMA | **40.78** |
| ARIMA | 148.86 | SCENT | 44.2 |
| N-BEATS | 106.41 | - | - |

## L. Computational Complexity Analysis

**Summary.** This section provides Big-O time comparisons between SCENT, AROMA, and FNO. In summary, SCENT's cost is sensitively affected by $S$ which denotes the number of tokens attended within sparse attention layers in CEN (Section 3.2) and CN (Section 3.4). Meanwhile, AROMA with a diffusion transformer backbone is costly if the number of refinement step $K$ and unrolling steps $T$ increase. Lastly, FNO is sensitive to the model width $C$ and also the unrolling steps $T$. We show that at the scale of the NS-3 experiment (Section 2), SCENT is the most expensive during training among three models, while the gap shrinks for larger unrolling steps. SCENT scales linearly with $W$, while AROMA scales linearly with $K$ and $T$. Depending on their values, one could be more expensive than the other. Notably, $W \approx \frac{T}{t_h}$ where $t_h$ is the time horizon of SCENT, thanks to Warp-Unrolling Forecasting (Section 3.3, Fig. 3). Thus, for a longer time forecasting SCENT is more efficient. Please refer to following subsections for big-O time complexity derivations for individual models.

*Table 5.* Big-O time complexity

| MODEL | FORMULA |
|---|---|
| FNO | $O(T(LN \log NC + LNd^2C))$ |
| AROMA | $O((2N + 4KTL_2M + L_1M)Md)$ |
| SCENT | $O(WL_sNSd + WNMd + WL_mM^2d)$ |

### L.1. Big-O time complexity of SCENT

For SCENT equipped with:

- **Sparse attention:** $L_s$ layers where each of the $N$ tokens attends to $S$ tokens ($S \ll N$).
- **Cross-attention:** Two layers between $N$ and $M$ tokens.
- **Self-attention:** $L_m$ layers operating on $M$ tokens.

*Table 6.* Hyperparameters derived from experiments in Section 4.2.

| PARAMETER | SYMBOL | VALUE |
|---|---|---|
| FOURIER LAYERS IN FNO | $L$ | 4 |
| SPATIAL GRID POINTS | $N$ | 2000 |
| FOURIER MODES | $d$ | 16 |
| MODEL WIDTH | $C$ | 60 |
| UNROLLING STEPS (FNO, AROMA) | $T$ | 1 OR 20 |
| UNROLLING STEPS (SCENT) | $W$ | 1 OR 7 |
| SPARSE ATTENTION LAYERS | $L_s$ | 6 |
| SELF-ATTENTION LAYERS | $L_m$ | 2 |
| SPARSE ATTENTION TOKENS | $S$ | 500 |
| COMPRESSED TOKENS | $M$ | 128 |
| LATENT CHANNELS | $d$ | 128 |

*Table 7.* Sample cost computed using hyperparameters in Table 6

| | UNROLLING $T = 1$ | | UNROLLING $T = 20$ | |
| MODEL | COST | RELATIVE SCALE | COST | RELATIVE SCALE |
|---|---|---|---|---|
| FNO | 1.28E+08 | 1.0 | 2.56E+09 | 1.0 |
| AROMA | 2.29E+08 | 1.78 | 3.09E+09 | 1.21 |
| SCENT | 8.04E+08 | 6.28 | 5.63E+09 | 2.20 |

### Sparse Attention Blocks

Each token from $N$ attends to only $S$ tokens, reducing the full self-attention cost from $O(N^2 d)$ to:

$$O(L_s NSd)$$

### Cross-Attention Between $N$ and $M$ Tokens

Each cross-attention operation has cost:

$$O(NMd)$$

With two such layers, the total remains:

$$O(NMd)$$

### Self-Attention on $M$ Tokens

A self-attention block on $M$ tokens with $L_m$ layers incurs:

$$O(L_m M^2 d)$$

### Final Big-O Complexity

Summing all contributions:

$$O(L_s NSd + NMd + L_m M^2 d)$$

Considering $W$ unrolling during inference, it becomes:

$$O(WL_s NSd + WNMd + WL_m M^2 d)$$

### L.2. Big-O for AROMA

AROMA's computational complexity is provided as following (Serrano et al., 2024):

$$O\big((2N + 4KL_2 M + L_1 M)Md\big) \tag{4}$$

where:

- $K$ = Number of refinement steps
- $T$ = Number of autoregressive calls in unrolling
- $L_2, L_1$ = Number of layers in different parts of the architecture
- $N$ = Number of observations
- $M$ = Number of tokens used to compress information
- $d$ = Number of channels used in the attention mechanism

Diffusion Transformer incurs additional costs due to iterative refinement:

$$O(4KTL_2M^2d) \tag{5}$$

which scales with $K$ (refinement steps) and $T$ (autoregressive calls).

## L.3. Big-O for FNO

For an FNO (Li et al., 2020) model with:

- $L$ = Number of Fourier layers
- $N$ = Number of spatial grid points
- $d$ = Number of Fourier modes (spectral channels)
- $C$ = Model width (number of feature channels per layer)
- $T$ = Number of autoregressive unrolling

The key computational operations are:

**Fast Fourier Transform (FFT) and Inverse FFT (IFFT)**

$$O(LN \log NC) \tag{6}$$

FFT is applied across all feature channels.

**Linear Transform in Fourier Space**

$$O(LNd^2C) \tag{7}$$

Applies transformations to Fourier modes across all channels.

**Total Complexity for Multiple Layers**

$$O\left(T(LN \log NC + LNd^2C)\right) \tag{8}$$

The unrolling factor $T$ accounts for repeated forward passes in autoregressive prediction.

