# OpenReview forum: "SCENT: Robust Spatiotemporal Learning for Continuous Scientific Data via Scalable Conditioned Neural Fields"
_ICML.cc/2025/Conference — ICML 2025 poster_

### Official Review · Reviewer_9YaB · 2025-03-12

**Overall Recommendation:** 3

**Summary:**

This paper presents SCENT, which is a scalable and continuity-informed spatiotemporal learning framework designed to model complex scientific data. Using a transformer-based architecture with learnable queries and sparse attention, it unifies interpolation, reconstruction, and forecasting. Extensive experiments demonstrate its satisfied performance across various datasets, offering superior scalability and robustness against sparse and noisy data.

## Update after rebuttal:
The authors addressed the main concerns I raised, including the addition of RainNet experiments and comparisons with STFNN. While full statistical significance testing is still limited, I appreciate the substantial effort made to strengthen empirical validation. I have updated my score to 3 accordingly.

**Claims And Evidence:**

The paper claims that SCENT is scalable and computationally efficient. However, while the model appears manageable for small values of M, its efficiency for large-scale datasets remains uncertain. How is M determined in practice? Different values of M may significantly impact the cross-attention mechanism, yet there are no ablation studies provided to support this claim. Adding such an analysis would strengthen the evidence for scalability and efficiency.

**Essential References Not Discussed:**

STFNN [3] also utilizes INRs and focuses on unified spatiotemporal modeling. A direct comparison between SCENT and STFNN would help clarify their differences and contributions.

**Experimental Designs Or Analyses:**

1.	The paper does not include several closely related methods. Specifically, it lacks comparisons with ST grid forecasting models, such as AutoST [2] and ST-ResNet [3], as well as ST field-based methods, such as STFNN [4].
2.	The reported experimental results (Table 1 and 2) lack standard deviations and statistical significance markers (e.g., confidence intervals).

Reference

[2] Autost: Efficient neural architecture search for spatio-temporal prediction. KDD. 2020.

[3]  Deep Spatio-Temporal Residual Networks for Citywide Crowd Flows Prediction. AAAI 2017.

[4] Spatio-Temporal Field Neural Networks for Air Quality Inference. IJCAI 2024.

**Methods And Evaluation Criteria:**

The study does not include additional real-world datasets across different domains to better demonstrate the model’s generalizability. For example, RainNet [1].

[1] Rainnet v1. 0: a convolutional neural network for radar-based precipitation nowcasting. Geoscientific Model Development. 2020

**Other Comments Or Suggestions:**

I noticed that the related work is included in the appendix. I recommend incorporating it into the main text to ensure a more comprehensive and cohesive presentation.

**Other Strengths And Weaknesses:**

Strengths: The paper includes clear visualization.

Weakness: See previous section.

**Questions For Authors:**

See previous section.

**Relation To Broader Scientific Literature:**

The key contributions of this paper align with and extend multiple areas of ST learning, INRs, and scalable deep learning models. Compared to FNOs, it generalizes better to real-world data but needs validation on diverse domains like RainNet and include a comparisons with STFNN.

**Theoretical Claims:**

The paper does not introduce new theoretical results.

---

> ### Author Rebuttal · Authors · 2025-04-01
>
> We sincerely thank the reviewer for the insightful discussions, suggested papers, and constructive comments. It was a pleasant surprise to find substantial similarities as well as subtle yet important distinctions between SCENT, STFNN, and the referenced works. We found STFNN's inference mechanism and gradient-based formulation particularly innovative, and have therefore reached out to the authors for further exploration. As suggested, we will move the related work section to the main manuscript and provide more in-depth discussions on spatiotemporal forecasting methods, such as AutoST and ST-ResNet.
>
> $\ $
>
> **1. Comparing SCENT and STFNN**
>
> **Motivation and Similarities**. SCENT aims to develop a flexible model capable of generating continuous spatiotemporal fields from sparse observations, addressing the common scenario in scientific domains where sensor coverage is limited. Similarly, STFNN effectively infers unobserved regions using its sophisticated Pyramidal Inference. Both methods assume continuous fields and handle irregular, sparse, and noisy observations robustly.
>
> **Fundamental Differences in Generalization**. However, SCENT explicitly models a family of functions, acting as a generalizable implicit neural representation (GINR). It can represent various spatiotemporal scenarios conditioned on input data from different contexts. In contrast, STFNN models one specific spatiotemporal field (e.g., PM2.5 over China), allowing interpolation and extrapolation strictly within that field, rather than generalizing across distinct fields.
>
> **Architectural Design**. Both models utilize INRs but differ in structure and intent. SCENT employs an encoder–decoder architecture with attention-based querying to generalize across tasks. STFNN is more akin to a per-task INR, enhanced with gradient-based modeling and local graph-based correction to capture detailed local variations within a single domain.
>
> $\ $
>
> **2. RainNet: Rainfall Nowcasting**
>
> We appreciate for the suggestion. We have strengthened our experimental evaluation by incorporating RainNet and the new nowcasting dataset.
>
> **Dataset & Task**. We use the RY product from the German Weather Service (DWD), a quality-controlled rainfall composite at 1km~$\times$~1km spatial and 5-minute temporal resolution. Data from 2012-2016 are used for training, and 2017 for testing. The task is to predict rainfall fields for future timestamps $t \in [5, 10, \dots, 60]$ minutes, given four historical fields. Following the official preprocessing, we use 173,345 / 43,456 training / test instances, and downsample the original 900~$\times$~900 resolution to 64~$\times$~64 for faster training during the rebuttal period. SCENT is trained with a forecast horizon ($t_h$ = 60).
>
> **Results.** We report root mean-squared error (RMSE [mm$~h^{-1}$]) in the table below. SCENT consistently outperforms RainNet across all lead times, reducing MSE by approximately 30%. We attribute this improvement in part to SCENT's ability to train with variable target times $t_o$, which serves as a form of data augmentation.
>
>
> | Method|#Params|||||Lead|time|(mins)||||||
> |--|:--:|:--:|:--:|:--:|:--:|:--:|:--:|:--:|:--:|:--:|:--:|:--:|:--:|
> |||**5** |**10**|**15**|**20**| **25**|**30**| **35**| **40**|**45**|**50**|**55**|**60**|
> | RainNet|1.93M|0.445|0.431|0.439|0.460|0.491|0.526 |0.559|0.589|0.612 |0.628 | 0.639 | 0.646 |
> | SCENT|3.87M|0.319|0.341|0.354|0.366|0.377|0.389|0.399|0.409|0.417 | 0.425 | 0.434 | 0.440 |
> | Improvement |-| $28.3$% |$20.9$%|$19.4$%|$20.4$%|$26.0$%|$28.6$%|$30.6$% | $31.9$%|$31.9$%|$32.3$%|$32.1$% | $31.9$% |
> | | | | | | | | | | | | Metric:|RMSE|(mm$~h^{-1}$)|
>
> $\ $
>
> **3. Choosing M**
>
>  We agree with the reviewer that selecting the OPTIMAL number of latent tokens $M$ is important. While performance generally improves with larger $M$, we observe diminishing returns beyond $M=192$ on the S5 dataset—a saturation pattern also seen in Perceiver IO. This suggests that excessively large $M$ offers limited benefit while incurring higher computational cost, and a moderate $M$ can strike a better balance between performance and efficiency.
>
> | **M**|$\ $32|$\ $64|$\ $96|$\ $128|$\ $192|$\ $256|
> |--|:--:|:--:|:--:|:--:|:--:|:--:|
> |**Rel-MSE**|0.460|0.433|$\underline{0.422}$|0.425|**0.400**|0.427|
>
> $\ $
>
> **4. On statistical testing**
>
> We appreciate the reviewer’s suggestion. While full significance testing is infeasible due to time constraints, we provide evidence of robustness through multiple runs: for example, our small model (Rel-MSE = 0.467) shows a standard deviation of only 0.0042 across five seeds. Our evaluation also spans multiple benchmarks and a range of lead times, demonstrating consistent performance across diverse tasks and forecasting horizons. We acknowledge the value of statistical testing and will consider incorporating it in future revision.
>
> $\ $
>
> **We hope our responses address your concerns, and we’d be grateful if you’d consider updating your score accordingly.**

---

### Official Review · Reviewer_RkwD · 2025-03-12

**Overall Recommendation:** 4

**Summary:**

The authors introduce a new model called SCENT for spatiotemporal modelling such as for differential equations like Navier-stokes. This model can take irregular input data and generate outputs at arbitrary locations and times, and so is capable of forecasting and spatial interpolation. This model has an encoder - processor - decoder architecture using latent tokens in the processing layers. The authors add several new components to this architecture such as sparse self attention layers during encoding and decoding, and providing the required output time during input and output, limiting the number of recurrent steps required for forecasting. This model is compared to many other current methods on simulated and real world data and shows good performance. The authors show that this model scales well with model and dataset size.

**Claims And Evidence:**

Yes the claims are supported by clear and convincing evidence.

**Essential References Not Discussed:**

The FNO ("Fourier Neural Operator for Parametric Partial Differential Equations" (ICLR 2021)) seems to be one of the most effective methods compared to SCENT even though it requires regular gridded data. The authors adapt this method to work with the less regular data that is used in this work by padding out the grid, and find that it is often quite competitive with their approach. However there have been works that build on FNO to remove this dependency to gridded data (Geo-FNO from "Fourier Neural Operator with Learned Deformations for PDEs on General Geometries") and additionally provide further improvements (F-FNO from "Factorized Fourier Neural Operators" (ICLR 2023)). It would have been enlightening to cite and compare to one or both of these works or discuss why they are not suitable for the datasets used in this work.

**Experimental Designs Or Analyses:**

I investigated all of the experimental designs and analyses. I see from appendix C and D that some hyperparameter tuning was done for SCENT for the various different datasets / tasks. I would be interested if the same level of tuning / searching was performed for the various other approaches that are compared against. If not then some of the comparisons could be a bit unfair.

**Methods And Evaluation Criteria:**

The proposed datasets and evaluation metrics make sense, following those used in previous work. (e.g. "Fourier Neural Operator for Parametric Partial Differential Equations" (ICLR 2021) and "AirDelhi: Fine-Grained Spatio-Temporal Particulate Matter Dataset From Delhi For ML based Modeling" (NeurIPS 2023))

**Other Comments Or Suggestions:**

Some very minor issues I noticed:

Line 182 - right - latex formatting error

Line 326 - left - surely reconstruction is when delta t = 0?

Line 661 - treatSeach -> treats each

Figure 2 + table 3 - inconsistency in name of context encoding / embedding network

Table 4 - I am not sure why CORAL is more space and time continuous than the other approaches - it is still based on recursively using the processor module to step forward in time like the other approaches. OFormer and AROMA seem to be able to do basically the same things as CORAL here.

**Other Strengths And Weaknesses:**

I appreciate the effort the authors have gone through to compare to a wide variety of previous work in this area, and test them on both real world and synthetic datasets.

While the idea of essentially providing the required timestep at input and during decoding seems simple, it does not seem to be done in previous work and this could be quite helpful in preventing the accumulation of errors during recurrent processing steps usually required for forecasting. Vaughan et al. (2024) “Aardvark Weather: End-to-end data-driven weather prediction”, used for weather forecasting attempts to avoid this by training many different models, each for a different timestep, which feels less efficient than the approach used in this work.

The SCENT approach does show generally very good performance but in some cases it is only marginally better than alternative methods, and I wonder if these other approaches received the same level of hyperparameter optimization that is seen in the appendices for SCENT for each of the different datasets / tasks. If not then perhaps they would outperform SCENT with equivalent tuning.

I would have liked to see a comparison to newer FNO approaches considering that the basic version of this with a probably suboptimal approach to non regular data showed good performance.

**Questions For Authors:**

1. How did you select hyperparameters for the approaches you compared to? And what was your process for selecting these for your approach? Answering this will give me more confidence in how your approach performs compared to alternatives.

**Relation To Broader Scientific Literature:**

The SCENT architecture is quite similar to other approaches in the literature with a few key additions. The encoder - processor - decoder architecture with latent tokens is seen in CORAL ("Operator learning with neural fields: Tackling pdes on general geometries" (NeurIPS 2023)), AROMA ("AROMA: Preserving Spatial Structure for Latent PDE Modeling with Local Neural Fields" (NeurIPS 2024)), IPOT ("Inducing Point Operator Transformer: A Flexible and Scalable Architecture for Solving PDEs" (AAAI 2024)) for example.

The sparse attention layers in the encoder and decoder do not seem to be present in earlier works. “WUF” approach to limit recurrent time marching steps seems novel as well and could be effective in limiting the accumulation of errors.

The paper compares their results to a wide variety of works in the area, which is helpful and is sometimes not done in previous work, and must have required significant effort. They also make use of several previous datasets and introduce new datasets to provide a good comparison between these approaches.

**Theoretical Claims:**

Only real theoretical claims are the big O analyses in the appendices, which look right to me.

---

> ### Author Rebuttal · Authors · 2025-04-01
>
> **1. Hyperparameters**
> We appreciate the reviewer’s thoughtful questions regarding the extent of hyperparameter tuning conducted for SCENT in comparison to the baseline methods. This is indeed a crucial aspect when evaluating model performance fairly across methods. We included detailed hyperparameters in the appendix for transparency and reproducibility, and will release the code upon acceptance.
>
> **Inherited from Literature.** We clarify that we did not extensively tune hyperparameters for baseline methods, but instead adopted configurations from prior work. Many baselines, including IPOT and Perceiver-IO, share architectural similarities with our encoder-processor-decoder design, allowing us to inherit most of their recommended settings. For instance, Appendix Table C largely follows IPOT, with only minor changes such as warmup and total training steps.
>
> **Dataset Dependence.** Dataset-specific parameters (e.g., optimizer profiles, embedding configurations) reflect widely accepted strategies from prior literature, especially for common datasets like NS-3–5. These datasets exhibit varied dynamics but benefit from established training practices—such as long schedules for PDE dynamics and lower sensitivity to learning rate. As shown in Appendix Table D, most hyperparameters are consistent across datasets. We deliberately avoided aggressive tuning to promote generalizability. For example, embedding dimensions result from simple design choices—e.g., combining linear projections and Fourier features—rather than extensive tuning.
>
> **Baseline Hyperparameters.** When implementing the baselines, we made every effort to adhere to published or well-established hyperparameters. We first ensured each baseline was reproducible on its original dataset, then applied either the reported settings or those that worked well for SCENT—whichever yielded better performance. Given the complexity of our proposed data, which reflects realistic and large-scale scientific scenarios, we devoted substantial effort to adapting the baselines accordingly.
>
> **2. Newer FNO approaches**
> We thank the reviewer for highlighting recent FNO variants, including Geo-FNO and F-FNO. These works offer notable improvements, particularly in handling irregular geometries and sampling patterns. While we were not able to implement both approaches in full due to time constraints, we successfully implemented F-FNO on the S5 dataset and included those results in our evaluation. We evaluated two configurations of F-FNO with different parameter sizes (m1, m2). Notably, even the smaller variant (1M parameters) performs similarly to the standard FNO, demonstrating the efficiency of the architecture. When scaled to match the parameter count of SCENT and FNO (7.4M), F-FNO achieves significantly improved performance, approaching that of SCENT. We believe this provides meaningful insight into how newer FNO variants perform in challenging, non-grid-based scenarios.
>
> | Metric | SCENT | FNO  | F-FNO_m1 | F-FNO_m2 |
> |---|---|---|---|---|
> | # Params| 7.4M| 7.4M | 0.96M | 7.4M |
> | Rel-MSE| **0.326**| 0.377| 0.396| $\underline{0.347}$ |
>
> **3. On Vaughan et al. (2024) [1]**
> We appreciate the reviewer’s reference to the Aardvark Weather system. The idea of sequentially trained processors extending the recursive rollout strategy is particularly interesting. By allowing each processor to specialize in longer lead times, the approach may indeed improve inference stability and long-range forecasting performance. While the sequential training of multiple models could introduce computational overhead, the modular nature of the design offers flexibility that might benefit certain applications. Additionally, although the current formulation appears to target fixed-step forecasting, integrating mechanisms for continuous-time inference could be an exciting direction for future work. We see this as a promising and complementary line of research and welcome further discussion on how such ideas might be integrated with or compared to SCENT’s joint modeling capabilities.
>
> **4. On CORAL**
> As the reviewer correctly notes, OFormer, CORAL, and AROMA are all trained using recursive forecasting with a fixed time step. However, CORAL distinguishes itself by introducing a Neural ODE solver, $g_{\phi}$, as the autoregressive processor operating in the latent z-code space. As a result, CORAL adopts a two-stage training process: the first trains the autoencoder, and the second trains the Neural ODE for forecasting. The key advantage of this approach is that the learned ODE can be evaluated at any arbitrary time point, enabling CORAL to capture both spatial and temporal continuity.
>
> **We appreciate your thoughtful review and hope our clarifications address your questions.**
>
> [1] Allen, Anna, et al. "End-to-end data-driven weather prediction." *Nature* (2025): 1–3.

---

> > ### Comment · Reviewer_RkwD · 2025-04-07
> >
> > Thank you for your informative response. You have clarified most of the issues I had, and I appreciate the comparison to newer FNO approaches. I will increase my score to 4 in response.

---

### Official Review · Reviewer_WAR8 · 2025-03-13

**Overall Recommendation:** 2

**Summary:**

This paper introduces SCENT, a framework for spatiotemporal learning using Scalable Conditioned Neural Fields (CNFs). The model is built on a Transformer-based encoder-processor-decoder architecture, incorporating learnable queries and a query-wise cross-attention mechanism to capture multi-scale dependencies. A sparse attention mechanism is used to improve scalability.

**Claims And Evidence:**

Some claims lack sufficient supporting evidence:
1. Sparse Attention Justification – The paper claims sparse attention improves scalability, but does not compare against other sparse attention models (e.g., Longformer, Linformer).
2. Fourier Features Impact – SCENT uses Fourier features, but the paper does not analyze how different frequency bands affect prediction quality.

**Essential References Not Discussed:**

The paper does not cite or compare SCENT’s sparse attention to established models like Longformer, Linformer, and others, which are essential for evaluating its efficiency.

**Experimental Designs Or Analyses:**

The experimental design is generally sound, but there are some limitations:
1. The paper does not compare SCENT’s sparse attention to other sparse attention models (e.g., Longformer, Linformer), making it unclear how much it contributes to performance gains.
2. The paper does not analyze the impact of different frequency bands on spatial encoding, leaving a gap in understanding its effectiveness.

**Methods And Evaluation Criteria:**

1. Simulated datasets (Navier-Stokes, synthetic sensor data) are relevant, but real-world evaluation is limited (only AirDelhi). More diverse real-world datasets are needed.

**Other Comments Or Suggestions:**

Please refer to weaknesses.

**Other Strengths And Weaknesses:**

Pros:
1. The encoder-processor-decoder architecture is well-motivated, and the use of learnable queries and cross-attention improves scalability.
2. SCENT outperforms baselines like FNO, AROMA, CORAL, and OFormer in most scenarios.

Cons:
1. SCENT uses Fourier features for spatial encoding, but the impact of different frequency bands on prediction quality is not analyzed.
2. Most evaluations focus on simulated datasets (Navier-Stokes, synthetic sensor data). The AirDelhi dataset is the only real-world test, and results are not significantly better than AROMA.
3. The motivation and implementation details of the sparse attention mechanism in SCENT are unclear. The paper does not thoroughly explain why sparse attention is necessary beyond scalability and how it specifically enhances spatiotemporal learning. Additionally, it lacks a direct comparison with other sparse attention models, such as Longformer or Linformer, which could provide insight into the efficiency and effectiveness of SCENT’s attention mechanism.

**Questions For Authors:**

1. How do different frequency bands in Fourier feature encoding affect SCENT’s performance? An ablation study could strengthen the justification for their use.
2. Why is SCENT’s sparse attention mechanism chosen over existing methods like Longformer, or Linformer?
3. SCENT performs well on simulated datasets, but its AirDelhi results are not significantly better than AROMA. Can the model generalize effectively to other real-world datasets?

**Relation To Broader Scientific Literature:**

The paper builds on spatiotemporal learning, implicit neural representations (INRs), and conditioned neural fields (CNFs) but lacks engagement with key works. It does not compare SCENT’s sparse attention to models like Longformer, Linformer, nor analyze Fourier features’ impact.

**Theoretical Claims:**

The paper does not present formal theoretical proofs, but it makes implicit theoretical claims about the benefits of its architecture.

---

> ### Author Rebuttal · Authors · 2025-04-01
>
> We truly appreciate your valuable comments. We agree on your concerns and suggestions, hence provide below our thoughts and additional experimental results for each of the questions.
>
> $\ $
>
> **1. On Fourier feature**
>
> Although Fourier features are well established, their formulation can vary. Here, we describe our approach and present additional ablation studies. Let $R$ denote the maximum frequency resolution and $L$ the number of frequency bands. For the $i$th band, we define the frequency as
> $
> f_i = \frac{iR}{L}, \quad i=1,2,\ldots, L.
> $
> Then, the positional encoding for a scalar $x$ is given by the concatenation of sine and cosine functions:
> $
> \gamma(x)=\operatorname{concat}_{i=1}^L \Big[\sin\big(2\pi f_i x\big), \cos\big(2\pi f_i x\big)\Big].
> $
>
> We tune $R$ and $L$ to match the data’s inherent frequency characteristics: $R$ is set high enough to capture rapid variations, while $L$ is chosen to balance detail with computational efficiency. The following ablation experiments were conducted using a baseline SCENT model on dataset S5 with $M=32$ learnable queries, latent dimension $l=128$, and a batch size of 128.
>
> |# bands ($L$)|$\quad$4|$\quad$6|$\quad$8|$\quad$12|$\quad$16|
> |:---:|:---:|:---:|:---:|:---:|:---:|
> |**Rel-MSE**|$\underline{0.448}$|**0.446**|0.466|0.471|0.453|
>
>
> |Max resolution ($R$)|$\quad$ 5|$\quad$10|$\quad$20|$\quad$32|
> |:---:|:---:|:---:|:---:|:---:|
> | **Rel-MSE** | 0.514 |0.573| $\underline{0.471}$ | **0.443** |
>
> The results show that while the number of frequency bands $L$ has little impact on performance, a higher maximum resolution $R$ clearly improves it. Values were not explored beyond the Nyquist criterion (i.e., $R=32$).
>
> $\ $
>
> **2. SCENT’s sparse attention mechanism**
>
> We appreciate the reviewer's insightful question. Since scientific data typically appear as smooth and continuous signals, attending to all tokens may become redundant and inefficient for high sampling rates. In our approach, we employ random sparse attention for both the Context Embedding Network and Calibration Network (see Fig. 2), where each token attends to a random subset of $p$ tokens, reducing the complexity from $O(n^2)$ to $O(pn)$ with $p \ll n$. This simple random sparse attention mechanism has demonstrated strong empirical performance. We also acknowledge the extensive literature on efficient attention mechanisms – such as Longformer and Linformer – which may work equally well; indeed, additional experiments with these methods (with roughly matched parameter sizes for fair comparison) reveal that the Longformer (replacing sparse attentions within SCENT) outperforms the others, suggesting promising avenues for future research.
>
> | Method|Big-O Complexity|$\quad$ Variables/Notes|Rel-MSE
> |--|--|--|--|
> |Random Sparse Attention| $O(n·p$)| $p$: # tokens each query attends to|0.471|
> |SCENT+Linformer [1]|$O(n·k$) |$k$: projected dimension ($k \ll n$, constant)|0.496|
> |SCENT+Longformer [2]|$O(n·(w+g)$)|$w$: sliding window size; $g$: # global tokens|**0.440**|
>
> $\ $
>
> **3. Additional real-world datasets**
>
> Thank you for raising concerns over limited real dataset employed. Here we report explore additional spatiotemporal datasets.
>
> (i) **Rainfall Nowcasting**: Using DWD radar rainfall data, SCENT nowcasts lead times from 5 to 60 minutes based on four consecutive 5-minute intervals. Compared to the CNN-based RainNet baseline, SCENT reduces the root-mean-squared error (RMSE) by approximately 30\%. The table summarizes the performance metrics (RMSE in mm$~h^{-1}$) and the number of parameters for each method.
>
> | Method | # Params ||Lead|time|(mins)||
> |--|:---:|:--:|:--:|:--:|:--:|:--:|
> |||**5**|**10**|**30**|**60**|
> |RainNet [3]|1.93M|0.445|0.431|0.526|0.646|
> |SCENT|3.87M|**0.319**|**0.341**|**0.389**|**0.440**|
> ||||Metric:|RMSE|(mm$~h^{-1}$)||||
>
>
> (ii) **Kuroshio Path Prediction**: Using 50 years of CORA [4] reanalysis data, SCENT predicts the Kuroshio current's latitude over a 120-day horizon (with data from 1958–1997 for training and 1998–2007 for testing). SCENT demonstrates stable performance beyond a 50-day lead time. Table below shows that SCENT achieves lower RMSE (in degrees) compared to an LSTM baseline, particularly for longer forecast horizons.
>
> |Model|||Lead|Time|(days)|
> |--|--|:--:|:--:|:--:|:--:|
> |||10|30|60|120|
>  |LSTM||0.403|0.511|0.591|0.647|
>  |SCENT||**0.354**|**0.360**|**0.391**|**0.430**|
> |||||Metric:|RMSE (°)|
>
> **We hope our responses have addressed your key concerns and would be grateful if you would consider a higher score in light of these clarifications.**
>
> [1] Wang, S, et al. "Linformer: Self-attention with linear complexity", *arXiv* 2020.
>
> [2] Beltagy, I, et al. "Longformer: The long-document transformer", *arXiv* 2020.
>
> [3] Ayzel, G, et al. "RainNet v1.0: a convolutional neural network for radar-based precipitation nowcasting", *GMD* (2020).
>
> [4] Han, G. et al. "A new version of regional ocean reanalysis for coastal waters of China and adjacent seas", *Adv. Atmos. Sci.* (2013).

---

### Official Review · Reviewer_ujM5 · 2025-03-18

**Overall Recommendation:** 3

**Summary:**

This paper addresses common issues in scientific data, such as sparsity, noise, and multi-scale problems, by proposing a method called SCENT (Scalable Conditioned Neural Field) that can handle various spatio-temporal learning tasks like interpolation, reconstruction, and prediction. The paper is well-structured with clear motivation, and extensive experiments on both synthetic and real-world data demonstrate the method's effectiveness and scalability.

**Claims And Evidence:**

yes

**Essential References Not Discussed:**

yes

**Ethical Review Flag:**

Flag this paper for an ethics review.

**Ethics Expertise Needed:**

["Other expertise"]

**Experimental Designs Or Analyses:**

yes

**Methods And Evaluation Criteria:**

yes

**Other Comments Or Suggestions:**

No

**Other Strengths And Weaknesses:**

Weaknesses: I think the research scenario is too idealized. It could be worth considering real-world scenarios, like observational data from the Kuroshio, for example.

**Questions For Authors:**

No

**Relation To Broader Scientific Literature:**

yes

**Theoretical Claims:**

yes

---

> ### Author Rebuttal · Authors · 2025-04-01
>
> We appreciate the reviewer's suggestion. We agree that the Kuroshio current provides an excellent yet challenging testbed for evaluating SCENT. We use 50‐year records from the China Ocean Reanalysis (CORA) [1] as our benchmark and follow the data processing guidelines established by Wu et al. (2023) [2].
>
>
> **Background.** The Kuroshio current, originating from the North Equatorial Current (NEC) and flowing northward along the eastern side of the Philippine Islands, is the world's second-largest warm current. Accurately predicting its path is crucial because its variations significantly affect the exchange of water masses and heat between the North Pacific subtropical and subarctic circulations. CORA provides daily oceanographic reanalysis data for the Kuroshio current spanning 50 years (January 1958-December 2007).
>
> **Implementation.** We adopt the four baseline methods from Wu et al. (2023) for comparison. We perform a 120-day prediction experiment, using data from the first 40 years (1958-1997) for training and the final 10 years (1998-2007) for testing. During training, SCENT is used to predict the Kuroshio path in terms of latitude (ranging from 29°N to 36°N) for various forecast horizons ($t_o$). In the test phase, we measure the root mean-squared error (RMSE, in degrees) against the true latitude for every $t_o \in [1, 120]$ days, and we present comparisons at 10-day intervals.
>
>
> **Results.** Comparison analyses are shown below. We categorize baseline methods into those employing feature engineering (FE) and those that are purely neural network–based. LSTM is used as the baseline, and we also compare FE-enhanced LSTM variants, where FE includes empirical orthogonal functions (EOF) and complete ensemble empirical mode decomposition with adaptive noise (CEEMDAN). The CEEMDAN-based variants perform competitively across all lead times, while SCENT exhibits second-best performance at and beyond a 50-day lead time. The relative strength of the FE methods may be due to the limited data scale—with 14,610 instances in the training split and 3,652 in the test split. However, our scalability study (Fig. 4) suggests that SCENT could outperform the FE methods when larger datasets become available. Additionally, SCENT maintains stable performance even as the lead time increases. This is in stark contrast to the other baselines, which experience significantly sharper performance degradation with longer lead times.
>
>
> | Type  | Model            |                   |                                                   |       |       |       |    Lead    |   Time     |   (days)    |       |       |       |     |     |
> |------------------|------------------|-------------------|:-----------------------------------------------------------------:|:-----:|:-----:|:-----:|:-----:|:-----:|:-----:|:-----:|:-----:|:-----:|:-----:|:-----:|
> |            |                  |                   | 10 | 20 | 30 | 40 | 50 | 60 | 70 | 80 | 90 | 100| 110| 120|
>  | FE+NN | EOF_LSTM         |                   | 0.391  | 0.448  | 0.490  | 0.518  | 0.546  | 0.570  | 0.592  | 0.601  | 0.612  | 0.619  | 0.627  | 0.628  |
>  | FE+NN| CEEMDAN_LSTM     |                   | $\underline{0.243}$  | $\underline{0.294}$  | $\underline{0.328}$  | $\underline{0.357}$  | 0.380  | 0.399  | 0.417  | 0.432  | 0.446  | 0.464  | 0.481  | 0.493  |
>  | FE+NN| EOF_SEEMDAN_LSTM |                   | **0.176**  | **0.229**  | **0.262**  | **0.279**  | **0.303**  | **0.325**  | **0.337**  | **0.346**  | **0.357**  | **0.371**  | **0.386**  | **0.399**  |
>  | Pure NN| LSTM             |                   | 0.403  | 0.468  | 0.511  | 0.546  | 0.576  | 0.591  | 0.605  | 0.617  | 0.628  | 0.643  | 0.646  | 0.647  |
>  | Pure NN| SCENT            |                   | 0.354  | 0.348  | 0.360  | 0.365  | $\underline{0.374}$  | $\underline{0.391}$  | $\underline{0.390}$  | $\underline{0.408}$ | $\underline{0.410}$  | $\underline{0.419}$ | $\underline{0.415}$  | $\underline{0.430}$  |
> |            |                  |                   |  | | | | ||||| | Metric: | RMSE (°)  |
>
>
> **Thank you again for your time and feedback. We respectfully ask you to consider a revised score if you feel our responses address your concerns.**
>
> [1] Han, G. et al. "A new version of regional ocean reanalysis for coastal waters of China and adjacent seas", *Adv. Atmos. Sci*. 30, 974–982 (2013).
>
>
> [2] Wu, X. et al. "Deep Learning–Based Prediction of Kuroshio Path South of Japan", *J. Atmos. Oceanic Tech*. 40.2 (2023).

---

### Decision · Program_Chairs · 2025-05-01

**Decision:**

Accept (poster)

**Comment:**

The proposed method achieves state-of-the-art performance on several benchmarks. The reviewers had doubts about the limited real-world benchmarks to evaluate the method. In the rebuttal authors provided in-depth clarifications and several additional experimental results that support the efficacy of the method.
I lean towards acceptance.